# Kernel quadrature with DPPs

**Ayoub Belhadji,   Rémi Bardenet,   Pierre Chainais**
Univ. Lille, CNRS, Centrale Lille, UMR 9189 - CRIStAL, Villeneuve d'Ascq, France
{ayoub.belhadji, remi.bardenet, pierre.chainais}@univ-lille.fr

## Abstract

We study quadrature rules for functions from an RKHS, using nodes sampled from a determinantal point process (DPP). DPPs are parametrized by a kernel, and we use a truncated and saturated version of the RKHS kernel. This link between the two kernels, along with DPP machinery, leads to relatively tight bounds on the quadrature error, that depends on the spectrum of the RKHS kernel. Finally, we experimentally compare DPPs to existing kernel-based quadratures such as herding, Bayesian quadrature, or leverage score sampling. Numerical results confirm the interest of DPPs, and even suggest faster rates than our bounds in particular cases.

## 1   Introduction

Numerical integration [11] is an important tool for Bayesian methods [38] and model-based machine learning [32]. Formally, numerical integration consists in approximating

$$\int_{\mathcal{X}} f(x)g(x)\mathrm{d}\omega(x) \approx \sum_{j\in[N]} w_j f(x_j), \tag{1}$$

where $\mathcal{X}$ is a topological space, $\mathrm{d}\omega$ is a Borel probability measure on $\mathcal{X}$, $g$ is a square integrable function, and $f$ is a function belonging to a space to be precised. In the quadrature formula (1), the $N$ points $x_1, \ldots, x_N \in \mathcal{X}$ are called the quadrature *nodes*, and $w_1, \ldots, w_N$ the corresponding *weights*.

The accuracy of a quadrature rule is assessed by the quadrature error, i.e., the absolute difference between the left-hand side and the right-hand side of (1). Classical Monte Carlo algorithms, like importance sampling or Markov chain Monte Carlo [39], pick up the nodes as either independent samples or a sample from a Markov chain on $\mathcal{X}$, and all achieve a root mean square quadrature error in $\mathcal{O}(1/\sqrt{N})$. Quasi-Monte Carlo quadrature [12] is based on deterministic, low-discrepancy sequences of nodes, and typical error rates for $\mathcal{X} = \mathbb{R}^d$ are $\mathcal{O}(\log^d N/N)$. Recently, kernels have been used to derive quadrature rules such as herding [2, 9], Bayesian quadrature [19, 35], sophisticated control variates [28, 33], and leverage-score quadrature [1] under the assumption that $f$ lies in a RKHS. The main theoretical advantage is that the resulting error rates are faster than classical Monte Carlo and adapt to the smoothness of $f$.

In this paper, we propose a new quadrature rule for functions in a given RKHS. Our nearest scientific neighbour is [1], but instead of sampling nodes independently, we leverage dependence and use a repulsive distribution called a projection determinantal point process (DPP), while the weights are obtained through a simple quadratic optimization problem. DPPs were originally introduced by [29] as probabilistic models for beams of fermions in quantum optics. Since then, DPPs have been thoroughly studied in random matrix theory [21], and have more recently been adopted in machine learning [26] and Monte Carlo methods [3].

In practice, a projection DPP is defined through a reference measure $\mathrm{d}\omega$ and a repulsion kernel $\mathfrak{K}$. In our approach, the repulsion kernel is a modification of the underlying RKHS kernel. This ensures that sampling is tractable, and, as we shall see, that the expected value of the quadrature error is

controlled by the decay of the eigenvalues of the integration operator associated to the measure $\mathrm{d}\omega$. Note that quadratures based on projection DPPs have already been studied in the literature: implicitly in [22, Corollary 2.3] in the simple case where $\mathcal{X} = [0, 1]$ and $\mathrm{d}\omega$ is the uniform measure, and in [3] for $[0, 1]^d$ and more general measures. In the latter case, the quadrature error is asymptotically of order $N^{-1/2-1/2d}$ [3], with $f$ essentially $\mathcal{C}^1$. In the current paper, we leverage the smoothness of the integrand to improve the convergence rate of the quadrature in general spaces $\mathcal{X}$.

This article is organized as follows. Section 2 reviews kernel-based quadrature. In Section 3, we recall some basic properties of projection DPPs. Section 4 is devoted to the exposition of our main result, along with a sketch of proof. We give precise pointers to the supplementary material for missing details. Finally, in Section 5 we illustrate our result and compare to related work using numerical simulations, for the uniform measure in $d = 1$ and 2, and the Gaussian measure on $\mathbb{R}$.

**Notation.** Let $\mathcal{X}$ be a topological space equipped with a Borel measure $\mathrm{d}\omega$ and assume that the support of $\mathrm{d}\omega$ is $\mathcal{X}$. Let $\mathbb{L}_2(\mathrm{d}\omega)$ be the Hilbert space of square integrable, real-valued functions defined on $\mathcal{X}$, with the usual inner product denoted by $\langle \cdot, \cdot \rangle_{\mathrm{d}\omega}$, and the associated norm by $\|.\|_{\mathrm{d}\omega}$. Let $k : \mathcal{X} \times \mathcal{X} \to \mathbb{R}_+$ be a symmetric and continuous function such that, for any finite set of points in $\mathcal{X}$, the matrix of pairwise kernel evaluations is positive semi-definite. Denote by $\mathcal{F}$ the associated reproducing kernel Hilbert space (RKHS) of real-valued functions [5]. We assume that $x \mapsto k(x, x)$ is integrable with respect to the measure $\mathrm{d}\omega$ so that $\mathcal{F} \subset \mathbb{L}_2(\mathrm{d}\omega)$. Define the integral operator

$$\mathbf{\Sigma} f(\cdot) = \int_{\mathcal{X}} k(\cdot, y) f(y) \mathrm{d}\omega(y), \quad f \in \mathbb{L}_2(\mathrm{d}\omega). \tag{2}$$

By construction, $\mathbf{\Sigma}$ is self-adjoint, positive semi-definite, and trace-class [40]. For $m \in \mathbb{N}$, denote by $e_m$ the $m$-th eigenfunction of $\mathbf{\Sigma}$, normalized so that $\|e_m\|_{\mathrm{d}\omega} = 1$ and $\sigma_m$ the corresponding eigenvalue. The integrability of the diagonal $x \mapsto k(x, x)$ implies that $\mathcal{F}$ is compactly embedded in $\mathbb{L}_2(\mathrm{d}\omega)$, that is, the identity map $I_{\mathcal{F}} : \mathcal{F} \longrightarrow \mathbb{L}_2(\mathrm{d}\omega)$ is compact; moreover, since $\mathrm{d}\omega$ is of full support in $\mathcal{X}$, $I_{\mathcal{F}}$ is injective [44]. This implies a Mercer-type decomposition of $k$,

$$k(x, y) = \sum_{m \in \mathbb{N}^*} \sigma_m e_m(x) e_m(y), \tag{3}$$

where $\mathbb{N}^* = \mathbb{N} \smallsetminus \{0\}$ and the convergence is point-wise [45]. Moreover, for $m \in \mathbb{N}^*$, we write $e_m^{\mathcal{F}} = \sqrt{\sigma_m} e_m$. Since $I_{\mathcal{F}}$ is injective [45], $(e_m^{\mathcal{F}})_{m \in \mathbb{N}^*}$ is an orthonormal basis of $\mathcal{F}$. Unless explicitly stated, we assume that $\mathcal{F}$ is dense in $\mathbb{L}_2(\mathrm{d}\omega)$, so that $(e_m)_{m \in \mathbb{N}^*}$ is an orthonormal basis of $\mathbb{L}_2(\mathrm{d}\omega)$. For more intuition, under these assumptions, $f \in \mathcal{F}$ if and only if $\sum_m \sigma_m^{-1} \langle f, e_m \rangle_{\mathbb{L}_2(\mathrm{d}\omega)}^2$ converges.

## 2 Related work on kernel-based quadrature

When the integrand $f$ belongs to the RKHS $\mathcal{F}$ of kernel $k$ [10], the quadrature error reads [41]

$$\int_{\mathcal{X}} f(x) g(x) \mathrm{d}\omega(x) - \sum_{j \in [N]} w_j f(x_j) = \langle f, \mu_g - \sum_{j \in [N]} w_j k(x_j, .) \rangle_{\mathcal{F}}$$

$$\leq \|f\|_{\mathcal{F}} \left\| \mu_g - \sum_{j \in [N]} w_j k(x_j, .) \right\|_{\mathcal{F}}, \tag{4}$$

where $\mu_g = \int_{\mathcal{X}} g(x) k(x, .) \mathrm{d}\omega(x)$ is the so-called *mean element* [13, 31]. A tight approximation of the mean element by a linear combination of functions $k(x_j, .)$ thus guarantees low quadrature error. The approaches described in this section differ by their choice of nodes and weights.

### 2.1 Bayesian quadrature and the design of nodes

*Bayesian Quadrature* initially [27] considered a fixed set of nodes and put a Gaussian process prior on the integrand $f$. Then, the weights were chosen to minimize the posterior variance of the integral of $f$. If the kernel of the Gaussian process is chosen to be $k$, this amounts to minimizing the RHS of (4). The case of the Gaussian reference measure was later investigated in detail [35], while parametric integrands were considered in [30]. Rates of convergence were provided in [8] for specific kernels on compact spaces, under a *fill-in* condition [47] that encapsulates that the nodes must progressively fill up the (compact) space.

Finding the weights that optimize the RHS of (4) for a fixed set of nodes is a relatively simple task, see later Section 4.1, the cost of which can even be reduced using symmetries of the set of nodes [20, 24]. Jointly optimizing on the nodes and weights, however, is only possible in specific cases [6, 23]. In general, this corresponds to a non-convex problem with many local minima [16, 34]. While [36] proposed to sample i.i.d. nodes from the reference measure $d\omega$, greedy minimization approaches have also been proposed [19, 34]. In particular, *kernel herding* [9] corresponds to uniform weights and greedily minimizing the RHS in (4). This leads to a fast rate in $\mathcal{O}(1/N)$, but only when the integrand is in a finite-dimensional RKHS. Kernel herding and similar forms of *sequential* Bayesian quadrature are actually linked to the Frank-Wolfe algorithm [2, 7, 19]. Beside the difficulty of proving fast convergence rates, these greedy approaches still require heuristics in practice.

## 2.2 Leverage-score quadrature

In [1], the author proposed to sample the nodes $(x_j)$ i.i.d. from some proposal distribution $q$, and then pick weights $\hat{w}$ in (1) that solve the optimization problem

$$\min_{w \in \mathbb{R}^N} \left\| \mu_g - \sum_{j \in [N]} \frac{w_j}{q(x_j)^{1/2}} k(x_j, .) \right\|_{\mathcal{F}}^2 + \lambda N \|w\|_2^2, \tag{5}$$

for some regularization parameter $\lambda > 0$. Proposition 1 gives a bound on the resulting approximation error of the mean element for a specific choice of proposal pdf, namely the leverage scores

$$q_\lambda^*(x) \propto \langle k(x,.), \boldsymbol{\Sigma}^{-1/2}(\boldsymbol{\Sigma} + \lambda \mathbb{I}_{\mathbb{L}_2(d\omega)})^{-1} \boldsymbol{\Sigma}^{-1/2} k(x,.) \rangle_{\mathbb{L}_2(d\omega)} = \sum_{m \in \mathbb{N}} \frac{\sigma_m}{\sigma_m + \lambda} e_m(x)^2. \tag{6}$$

**Proposition 1** (Proposition 2 in [1]). *Let* $\delta \in [0, 1]$, *and* $d_\lambda = \mathrm{Tr}\, \boldsymbol{\Sigma}(\boldsymbol{\Sigma} + \lambda \boldsymbol{I})^{-1}$. *Assume that* $N \geq 5 d_\lambda \log(16 d_\lambda/\delta)$, *then*

$$\mathbb{P}\left( \sup_{\|g\|_{d\omega} \leq 1} \inf_{\|\boldsymbol{w}\|^2 \leq \frac{4}{N}} \left\| \mu_g - \sum_{j \in [N]} \frac{w_j}{q_\lambda(x_j)^{1/2}} k(x_j, .) \right\|_{\mathcal{F}}^2 \leq 4\lambda \right) \geq 1 - \delta. \tag{7}$$

In other words, Proposition 1 gives a uniform control on the approximation error $\mu_g$ by the subspace spanned by the $k(x_j, .)$ for $g$ belonging to the unit ball of $\mathbb{L}_2(d\omega)$, where the $(x_j)$ are sampled i.i.d. from $q_\lambda^*$. The required number of nodes is equal to $\mathcal{O}(d_\lambda \log d_\lambda)$ for a given approximation error $\lambda$. However, for fixed $\lambda$, the approximation error in Proposition 1 does not go to zero when $N$ increases. One theoretical workaround is to make $\lambda = \lambda(N)$ decrease with $N$. However, the coupling of $N$ and $\lambda$ through $d_\lambda$ makes it very intricate to derive a convergence rate from Proposition 1. Moreover, the optimal density $q_\lambda^*$ is in general only available as the limit (6), which makes sampling and evaluation difficult. Finally, we note that Proposition 1 highlights the fundamental role played by the spectral decomposition of the operator $\boldsymbol{\Sigma}$ in designing and analyzing kernel quadrature rules.

## 3 Projection determinantal point processes

Let $N \in \mathbb{N}^*$ and $(\psi_n)_{n \in [N]}$ an orthonormal family of $\mathbb{L}_2(d\omega)$, and assume for simplicity that $\mathcal{X} \subset \mathbb{R}^d$ and that $d\omega$ has density $\omega$ with respect to the Lebesgue measure. Define the repulsion kernel

$$\mathfrak{K}(x, y) = \sum_{n \in [N]} \psi_n(x) \psi_n(y), \tag{8}$$

not to be mistaken for the RKHS kernel $k$. One can show [18, Lemma 21] that

$$\frac{1}{N!} \mathrm{Det}(\mathfrak{K}(x_i, x_j)_{i,j \in [N]}) \prod_{i \in [N]} \omega(x_i) \tag{9}$$

is a probability density over $\mathcal{X}^N$. When $x_1, \ldots, x_N$ have distribution (9), the set $\boldsymbol{x} = \{x_1, \ldots x_N\}$ is said to be a projection DPP[1] with reference measure $d\omega$ and kernel $\mathfrak{K}$. Note that the kernel $\mathfrak{K}$ is a positive definite kernel so that the determinant in (9) is non-negative. Equation (9) is key to

understanding DPPs. First, loosely speaking, the probability of seeing a point of $\boldsymbol{x}$ in an infinitesimal volume around $x_1$ is $\mathfrak{K}(x_1, x_1)\omega(x_1)\mathrm{d}x_1$. Note that when $d = 1$ and $(\psi_n)$ are the family of orthonormal polynomials with respect to $\mathrm{d}\omega$, this marginal probability is related to the optimal proposal $q_\lambda$ in Section 2.2; see Appendix E.2. Second, the probability of simultaneously seeing a point of $\boldsymbol{x}$ in an infinitesimal volume around $x_1$ and one around $x_2$ is

$$\left[\mathfrak{K}(x_1, x_1)\,\mathfrak{K}(x_2, x_2) - \mathfrak{K}(x_1, x_2)^2\right]\omega(x_1)\omega(x_2)\,\mathrm{d}x_1\mathrm{d}x_2$$
$$\leq [\mathfrak{K}(x_1, x_1)\omega(x_1)\mathrm{d}x_1]\,[\mathfrak{K}(x_2, x_2)\omega(x_2)\mathrm{d}x_2].$$

The probability of co-occurrence is thus always smaller than that of a Poisson process with the same intensity. In this sense, a projection DPP with symmetric kernel is a *repulsive* distribution, and $\mathfrak{K}$ encodes its repulsiveness.

One advantage of DPPs is that they can be sampled exactly. Because of the orthonormality of $(\psi_n)$, one can write the chain rule for (9); see [18]. Sampling each conditional in turn, using e.g. rejection sampling [39], then yields an exact sampling algorithm. Rejection sampling aside, the cost of this algorithm is cubic in $N$ without further assumptions on the kernel. Simplifying assumptions can take many forms. In particular, when $d = 1$, and $\omega$ is a Gaussian, gamma [14], or beta [25] pdf, and $(\psi_n)$ are the orthonormal polynomials with respect to $\omega$, the corresponding DPP can be sampled by tridiagonalizing a matrix with independent entries, which takes the cost to $\mathcal{O}(N^2)$ and bypasses the need for rejection sampling. For further information on DPPs see [21, 43].

## 4 Kernel quadrature with projection DPPs

We follow in the footsteps of [1], see Section 2.2, but using a projection DPP rather than independent sampling to obtain the nodes. In a nutshell, we consider nodes $(x_j)_{j\in[N]}$ that are drawn from the projection DPP with reference measure $\mathrm{d}\omega$ and repulsion kernel

$$\mathfrak{K}(x, y) = \sum_{n\in[N]} e_n(x)e_n(y), \tag{10}$$

where we recall that $(e_n)$ are the normalized eigenfunctions of the integral operator $\boldsymbol{\Sigma}$. The weights $\boldsymbol{w}$ are obtained by solving the optimization problem

$$\min_{w\in\mathbb{R}^N} \|\mu_g - \boldsymbol{\Phi}\boldsymbol{w}\|_{\mathcal{F}}^2, \tag{11}$$

where

$$\boldsymbol{\Phi} : (w_j)_{j\in[N]} \mapsto \sum_{j\in[N]} w_j k(x_j, .) \tag{12}$$

is the reconstruction operator[2]. In Section 4.1 we prove that (11) almost surely has a unique solution $\hat{\boldsymbol{w}}$ and state our main result, an upper bound on the expected approximation error $\|\mu_g - \boldsymbol{\Phi}\hat{\boldsymbol{w}}\|_{\mathcal{F}}^2$ under the proposed Projection DPP. Section 4.2 gives a sketch of the proof of this bound.

### 4.1 Main result

Assuming that nodes $(x_j)_{j\in[N]}$ are known, we first need to solve the optimization problem (11) that relates to problem (5) without regularization ($\lambda = 0$). Let $\boldsymbol{x} = (x_1, \ldots, x_N) \in \mathcal{X}^N$, then

$$\|\mu_g - \boldsymbol{\Phi}\boldsymbol{w}\|_{\mathcal{F}}^2 = \|\mu_g\|_{\mathcal{F}}^2 - 2\boldsymbol{w}^{\mathsf{T}}\mu_g(x_j)_{j\in[N]} + \boldsymbol{w}^{\mathsf{T}}\boldsymbol{K}(\boldsymbol{x})\boldsymbol{w}, \tag{13}$$

where $\boldsymbol{K}(\boldsymbol{x}) = (k(x_i, x_j))_{i,j\in[N]}$. The right-hand side of (13) is quadratic in $\boldsymbol{w}$, so that the optimization problem (11) admits a unique solution $\hat{\boldsymbol{w}}$ if and only if $\boldsymbol{K}(\boldsymbol{x})$ is invertible. In this case, the solution is given by $\hat{\boldsymbol{w}} = \boldsymbol{K}(\boldsymbol{x})^{-1}\mu_g(x_j)_{j\in[N]}$. A sufficient condition for the invertibility of $\boldsymbol{K}(\boldsymbol{x})$ is given in the following proposition.

**Proposition 2.** *Assume that the matrix* $\boldsymbol{E}(\boldsymbol{x}) = (e_i(x_j))_{i,j\in[N]}$ *is invertible, then* $\boldsymbol{K}(\boldsymbol{x})$ *is invertible.*

The proof of Proposition 2 is given in Appendix D.1. Since the pdf (9) of the projection DPP with kernel (10) is proportional to $\mathrm{Det}^2\,\boldsymbol{E}(\boldsymbol{x})$, the following corollary immediately follows.

**Corollary 1.** *Let $\boldsymbol{x} = \{x_1, \ldots, x_N\}$ be a projection DPP with reference measure $d\omega$ and kernel (10). Then $\boldsymbol{K}(\boldsymbol{x})$ is a.s. invertible, so that (11) has unique solution $\hat{\boldsymbol{w}} = \boldsymbol{K}(\boldsymbol{x})^{-1}\mu_g(x_j)_{j \in [N]}$ a.s.*

We now give our main result that uses nodes $(x_j)_{j \in [N]}$ drawn from a well-chosen projection DPP.

**Theorem 1.** *Let $\boldsymbol{x} = \{x_1, \ldots, x_N\}$ be a projection DPP with reference measure $d\omega$ and kernel (10). Let $\hat{\boldsymbol{w}}$ be the unique solution to (11) and define $\|g\|_{d\omega,1} = \sum\limits_{n \in [N]} |\langle e_n, g\rangle_{d\omega}|$. Assume that $\|g\|_{d\omega} \leq 1$ and define $r_N = \sum\limits_{m \geq N+1} \sigma_m$, then*

$$\mathbb{E}_{\mathrm{DPP}} \|\mu_g - \boldsymbol{\Phi}\hat{\boldsymbol{w}}\|_{\mathcal{F}}^2 \leq 2\sigma_{N+1} + 2\|g\|_{d\omega,1}^2 \left( Nr_N + \sum_{\ell=2}^N \frac{\sigma_1}{\ell!^2} \left(\frac{Nr_N}{\sigma_1}\right)^\ell \right). \tag{14}$$

In particular, if $Nr_N = o(1)$, then the right-hand side of (14) is $Nr_N + o(Nr_N)$. For example, take $\mathcal{X} = [0,1]$, $d\omega$ the uniform measure on $\mathcal{X}$, and $\mathcal{F}$ the $s$-Sobolev space, then $\sigma_m = m^{-2s}$ [5]. Now, if $s > 1$, the expected worst case quadrature error is bounded by $Nr_N = \mathcal{O}(N^{2-2s}) = o(1)$. Another example is the case of the Gaussian measure on $\mathcal{X} = \mathbb{R}$, with the Gaussian kernel. In this case $\sigma_m = \beta\alpha^m$ with $0 < \alpha < 1$ and $\beta > 0$ [37] so that $Nr_N = N\frac{\beta}{1-\alpha}\alpha^{N+1} = o(1)$.

We have assumed that $\mathcal{F}$ is dense in $\mathbb{L}_2(d\omega)$ but Theorem 1 is valid also when $\mathcal{F}$ is finite-dimensional. In this case, denote $N_0 = \dim\mathcal{F}$. Then, for $n > N_0$, $\sigma_n = 0$ and $r_{N_0} = 0$, so that (14) implies

$$\|\mu_g - \boldsymbol{\Phi}\hat{\boldsymbol{w}}\|_{\mathcal{F}}^2 = 0 \text{ a.s.} \tag{15}$$

This compares favourably with herding, for instance, which comes with a rate in $\mathcal{O}(\frac{1}{N})$ for the quadrature based on herding with uniform weights [2, 9].

The constant $\|g\|_{d\omega,1}$ in (14) is the $\ell_1$ norm of the coefficients of projection of $g$ onto $\mathrm{Span}(e_n)_{n \in [N]}$ in $\mathbb{L}_2(d\omega)$. For example, for $g = e_n$, $\|g\|_{d\omega,1} = 1$ if $n \in [N]$ and $\|g\|_{d\omega,1} = 0$ if $n \geq N+1$. In the worst case, $\|g\|_{d\omega,1} \leq \sqrt{N}\|g\|_{d\omega} \leq \sqrt{N}$. Thus, we can obtain a uniform bound for $\|g\|_{d\omega} \leq 1$ in the spirit of Proposition 1, but with a supplementary factor $N$ in the upper bound in (14).

## 4.2 Bounding the approximation error under the DPP

In this section, we give the skeleton of the proof of Theorem 1, referring to the appendices for technical details. The proof is in two steps. First, we give an upper bound for the approximation error $\|\mu_g - \boldsymbol{\Phi}\hat{\boldsymbol{w}}\|_{\mathcal{F}}^2$ that involves the maximal principal angle between the functional subspaces of $\mathcal{F}$

$$\mathcal{E}_N^{\mathcal{F}} = \mathrm{Span}(e_n^{\mathcal{F}})_{n \in [N]} \quad \text{and} \quad \mathcal{T}(\boldsymbol{x}) = \mathrm{Span}(k(x_j, .))_{j \in [N]}.$$

DPPs allow closed form expressions for the expectation of trigonometric functions of such angles; see [4] and Appendix E.1 for the geometric intuition behind the proof. The second step thus consists in developing the expectation of the bound under the DPP.

### 4.2.1 Bounding the approximation error using principal angles

Let $\boldsymbol{x} = (x_1, \ldots, x_N) \in \mathcal{X}^N$ be such that $\mathrm{Det}\,\boldsymbol{E}(\boldsymbol{x}) \neq 0$. By Proposition 2, $\boldsymbol{K}(\boldsymbol{x})$ is non singular and $\dim\mathcal{T}(\boldsymbol{x}) = N$. The optimal approximation error writes

$$\|\mu_g - \boldsymbol{\Phi}\hat{\boldsymbol{w}}\|_{\mathcal{F}}^2 = \|\mu_g - \boldsymbol{\Pi}_{\mathcal{T}(\boldsymbol{x})}\mu_g\|_{\mathcal{F}}^2, \tag{16}$$

where $\boldsymbol{\Pi}_{\mathcal{T}(\boldsymbol{x})} = \boldsymbol{\Phi}(\boldsymbol{\Phi}^*\boldsymbol{\Phi})^{-1}\boldsymbol{\Phi}^*$ is the orthogonal projection onto $\mathcal{T}(\boldsymbol{x})$ with $\boldsymbol{\Phi}^*$ the dual[3] of $\boldsymbol{\Phi}$.

In other words, (16) equates the approximation error to $\|\boldsymbol{\Pi}_{\mathcal{T}(\boldsymbol{x})^\perp}\mu_g\|_{\mathcal{F}}^2$, where $\boldsymbol{\Pi}_{\mathcal{T}(\boldsymbol{x})^\perp}$ is the orthogonal projection onto $\mathcal{T}(\boldsymbol{x})^\perp$. Now we have the following lemma.

**Lemma 1.** *Assume that $\|g\|_{d\omega} \leq 1$ then $\|\boldsymbol{\Sigma}^{-1/2}\mu_g\|_{\mathcal{F}} \leq 1$ and*

$$\|\boldsymbol{\Pi}_{\mathcal{T}(\boldsymbol{x})^\perp}\mu_g\|_{\mathcal{F}}^2 \leq 2\left(\sigma_{N+1} + \|g\|_{d\omega,1}^2 \max_{n \in [N]} \sigma_n\|\boldsymbol{\Pi}_{\mathcal{T}(\boldsymbol{x})^\perp}e_n^{\mathcal{F}}\|_{\mathcal{F}}^2\right). \tag{17}$$

Now, to upper bound the right-hand side of (17), we note that $\sigma_n \|\mathbf{\Pi}_{\mathcal{T}(\boldsymbol{x})^\perp} e_n^{\mathcal{F}}\|_{\mathcal{F}}^2$ is the product of two terms: $\sigma_n$ is a decreasing function of $n$ while $\|\mathbf{\Pi}_{\mathcal{T}(\boldsymbol{x})^\perp} e_n^{\mathcal{F}}\|_{\mathcal{F}}^2$ is the interpolation error of the eigenfunction $e_n^{\mathcal{F}}$, measured in the $\|.\|_{\mathcal{F}}$ norm. We can bound the latter interpolation error uniformly in $n \in [N]$ using the geometric notion of maximal principal angle between $\mathcal{T}(\boldsymbol{x})$ and $\mathcal{E}_N^{\mathcal{F}} = \text{Span}(e_n^{\mathcal{F}})_{n \in [N]}$. This maximal principal angle is defined through its cosine

$$\cos^2 \theta_N(\mathcal{T}(\boldsymbol{x}), \mathcal{E}_N^{\mathcal{F}}) = \inf_{\substack{\boldsymbol{u} \in \mathcal{T}(\boldsymbol{x}), \boldsymbol{v} \in \mathcal{E}_N^{\mathcal{F}} \\ \|u\|_{\mathcal{F}}=1, \|v\|_{\mathcal{F}}=1}} \langle \boldsymbol{u}, \boldsymbol{v} \rangle_{\mathcal{F}}. \tag{18}$$

Similarly, we can define the $N$ principal angles $\theta_n(\mathcal{T}(\boldsymbol{x}), \mathcal{E}_N^{\mathcal{F}}) \in \left[0, \frac{\pi}{2}\right]$ for $n \in [N]$ between the subspaces $\mathcal{E}_N^{\mathcal{F}}$ and $\mathcal{T}(\boldsymbol{x})$. These angles quantify the relative position of the two subspaces. See Appendix C.3 for more details about principal angles. Now, we have the following lemma.

**Lemma 2.** *Let $\boldsymbol{x} = (x_1, \ldots, x_N) \in \mathcal{X}^N$ such that $\text{Det}\, \boldsymbol{E}(\boldsymbol{x}) \neq 0$. Then*

$$\max_{n \in [N]} \|\mathbf{\Pi}_{\mathcal{T}(\boldsymbol{x})^\perp} e_n^{\mathcal{F}}\|_{\mathcal{F}}^2 \leq \frac{1}{\cos^2 \theta_N(\mathcal{T}(\boldsymbol{x}), \mathcal{E}_N^{\mathcal{F}})} - 1 \leq \prod_{n \in [N]} \frac{1}{\cos^2 \theta_n(\mathcal{T}(\boldsymbol{x}), \mathcal{E}_N^{\mathcal{F}})} - 1. \tag{19}$$

To sum up, we have so far bounded the approximation error by the geometric quantity in the right-hand side of (19). Where projection DPPs shine is in taking expectations of such geometric quantities.

### 4.2.2 Taking the expectation under the DPP

The analysis in Section 4.2.1 is valid whenever $\text{Det}\, \boldsymbol{E}(\boldsymbol{x}) \neq 0$. As seen in Corollary 1, this condition is satisfied almost surely when $\boldsymbol{x}$ is drawn from the projection DPP of Theorem 1. Furthermore, the expectation of the right-hand side of (19) can be written in terms of the eigenvalues of the kernel $k$.

**Proposition 3.** *Let $\boldsymbol{x}$ be a projection DPP with reference measure $d\omega$ and kernel (10). Then,*

$$\mathbb{E}_{\text{DPP}} \prod_{n \in [N]} \frac{1}{\cos^2 \theta_n \left(\mathcal{T}(\boldsymbol{x}), \mathcal{E}_N^{\mathcal{F}}\right)} = \sum_{\substack{T \subset \mathbb{N}^* \\ |T| = N}} \frac{\prod_{t \in T} \sigma_t}{\prod_{n \in [N]} \sigma_n}. \tag{20}$$

The bound of Proposition 3, once reported in Lemma 2 and Lemma 1, already yields Theorem 1 in the special case where $\sigma_1 = \cdots = \sigma_N$. This seems a very restrictive condition, but next Proposition 4 shows that we can always reduce the analysis to that case. In fact, let the kernel $\tilde{k}$ be defined by

$$\tilde{k}(x, y) = \sum_{n \in [N]} \sigma_1 e_n(x) e_n(y) + \sum_{n \geq N+1} \sigma_n e_n(x) e_n(y) = \sum_{n \in \mathbb{N}^*} \tilde{\sigma}_n e_n(x) e_n(y), \tag{21}$$

and let $\tilde{\mathcal{F}}$ be the corresponding RKHS. Then one has the following inequality.

**Proposition 4.** *Let $\tilde{\mathcal{T}}(\boldsymbol{x}) = \text{Span} \left(\tilde{k}(x_j, .)\right)_{j \in [N]}$ and $\mathbf{\Pi}_{\tilde{\mathcal{T}}(\boldsymbol{x})^\perp}$ the orthogonal projection onto $\tilde{\mathcal{T}}(\boldsymbol{x})^\perp$ in $(\tilde{\mathcal{F}}, \langle ., . \rangle_{\tilde{\mathcal{F}}})$. Then,*

$$\forall n \in [N], \ \sigma_n \|\mathbf{\Pi}_{\mathcal{T}(\boldsymbol{x})^\perp} e_n^{\mathcal{F}}\|_{\mathcal{F}}^2 \leq \sigma_1 \|\mathbf{\Pi}_{\tilde{\mathcal{T}}(\boldsymbol{x})^\perp} e_n^{\tilde{\mathcal{F}}}\|_{\tilde{\mathcal{F}}}^2. \tag{22}$$

Simply put, capping the first eigenvalues of $k$ yields a new kernel $\tilde{k}$ that captures the interaction between the terms $\sigma_n$ and $\|\mathbf{\Pi}_{\mathcal{T}(\boldsymbol{x})^\perp} e_n^{\mathcal{F}}\|_{\mathcal{F}}^2$ such that we only have to deal with the term $\|\mathbf{\Pi}_{\tilde{\mathcal{T}}(\boldsymbol{x})^\perp} e_n^{\tilde{\mathcal{F}}}\|_{\tilde{\mathcal{F}}}^2$. Combining Proposition 3 with Proposition 4 applied to the kernel $\tilde{k}$ yields Theorem 1.

### 4.3 Discussion

We have arbitrarily introduced a product in the right-hand side of (19), which is a rather loose majorization. Our motivation is that the expected value of this symmetric quantity is tractable under the DPP. Getting rid of the product could make the bound much tighter. Intuitively, taking the upper bound in (20) to the power $1/N$ results in a term in $\mathcal{O}(r_N)$ for the RKHS $\tilde{\mathcal{F}}$. Improving the bound in (20) would require a de-symmetrization by comparing the maximum of the $1/\cos^2 \theta_\ell(\mathcal{T}(\boldsymbol{x}), \mathcal{E}_N^{\mathcal{F}})$ to

their geometric mean. An easier route than de-symmetrization could be to replace the product in (19) by a sum, but this is beyond the scope of this article.

In comparison with [1], we emphasize that the dependence of our bound on the eigenvalues of the kernel $k$, via $r_N$, is explicit. This is in contrast with Proposition 1 that depends on the eigenvalues of $\boldsymbol{\Sigma}$ through the degree of freedom $d_\lambda$ so that the necessary number of samples $N$ diverges when $\lambda \to 0$. On the contrary, our quadrature requires a finite number of points for $\lambda = 0$. It would be interesting to extend the analysis of our quadrature in the regime $\lambda > 0$.

# 5 Numerical simulations

## 5.1 The periodic Sobolev space and the Korobov space

Let $\mathrm{d}\omega$ be the uniform measure on $\mathcal{X} = [0, 1]$, and let the RKHS kernel be [5]

$$k_s(x, y) = 1 + \sum_{m \in \mathbb{N}^*} \frac{1}{m^{2s}} \cos(2\pi m(x - y)),$$

so that $\mathcal{F} = \mathcal{F}_s$ is the Sobolev space of order $s$ on $[0, 1]$. Note that $k_s$ can be expressed in closed form using Bernoulli polynomials [46]. We take $g \equiv 1$ in (1), so that the mean element $\mu_g \equiv 1$. We compare the following algorithms: $(i)$ the quadrature rule DPPKQ we propose in Theorem 1, $(ii)$ the quadrature rule DPPUQ based on the same projection DPP but with uniform weights, implicitly studied in [22], $(iii)$ the kernel quadrature rule (5) of [1], which we denote LVSQ for *leverage score quadrature*, with regularization parameter $\lambda \in \{0, 0.1, 0.2\}$ (note that the optimal proposal is $q_\lambda^* \equiv 1$), $(iv)$ herding with uniform weights [2, 9], $(v)$ sequential Bayesian quadrature (SBQ) [19] with regularization to avoid numerical instability, and $(vi)$ Bayesian quadrature on the uniform grid (UGBQ). We take $N \in [5, 50]$. Figures 1a and 1b show log-log plots of the worst case quadrature error w.r.t. $N$, averaged over 50 samples for each point, for $s \in \{1, 3\}$.

We observe that the approximation errors of all first four quadratures converge to 0 with different rates. Both UGBQ and DPPKQ converge to 0 with a rate of $\mathcal{O}(N^{-2s})$, which indicates that our $\mathcal{O}(N^{2-2s})$ bound in Theorem 1 is not tight in the Sobolev case. Meanwhile, the rate of DPPUQ is $\mathcal{O}(N^{-2})$ across the three values of $s$: it does not adapt to the regularity of the integrands. This corresponds to the CLT proven in [22]. LVSQ without regularization converges to 0 slightly slower than $\mathcal{O}(N^{-2s})$. Augmenting $\lambda$ further slows down convergence. Herding converges at an empirical rate of $\mathcal{O}(N^{-2})$, which is faster than the rate $\mathcal{O}(N^{-1})$ predicted by the theoretical analysis in [2, 9]. SBQ is the only one that seems to plateau for $s = 3$, although it consistently has the best performance for low $N$. Overall, in the Sobolev case, DPPKQ and UGBQ have the best convergence rate. UGBQ – known to be optimal in this case [6] – has a better constant.

Now, for a multidimensional example, consider the "Korobov" kernel $k_s$ defined on $[0, 1]^d$ by

$$\forall \boldsymbol{x}, \boldsymbol{y} \in [0, 1]^d, \ \ k_{s,d}(\boldsymbol{x}, \boldsymbol{y}) = \prod_{i \in [d]} k_s(x_i, y_i). \tag{23}$$

We still take $g \equiv 1$ in (1) so that $\mu_g \equiv 1$. We compare $(i)$ our DPPKQ, $(ii)$ LVSQ without regularization ($\lambda = 0$), $(iii)$ the kernel quadrature based on the uniform grid UGBQ, $(iv)$ the kernel quadrature SGBQ based on the sparse grid from [42], $(v)$ the kernel quadrature based on the Halton sequence HaltonBQ [15]. We take $N \in [5, 1000]$ and $s = 1$. The results are shown in Figure 1c. This time, UGBQ suffers from the dimension with a rate in $\mathcal{O}(N^{-2s/d})$, while DPPKQ, HaltonBQ and LVSQ ($\lambda = 0$) all perform similarly well. They scale as $\mathcal{O}((\log N)^{2s(d-1)} N^{-2s})$, which is a tight upper bound on $\sigma_{N+1}$, see [1] and Appendix B. SGBQ seems to lag slightly behind with a rate $\mathcal{O}((\log N)^{2(s+1)(d-1)} N^{-2s})$ [17, 42].

## 5.2 The Gaussian kernel

We now consider $\mathrm{d}\omega$ to be the Gaussian measure on $\mathcal{X} = \mathbb{R}$ along with the RKHS kernel $k_\gamma(x, y) = \exp[-(x - y)^2/2\gamma^2]$, and again $g \equiv 1$. Figure 1d compares the empirical performance of DPPKQ to the theoretical bound of Theorem 1, herding, crude Monte Carlo with i.i.d. sampling from $\mathrm{d}\omega$, and sequential Bayesian Quadrature, where we again average over 50 samples. We take $N \in [5, 50]$

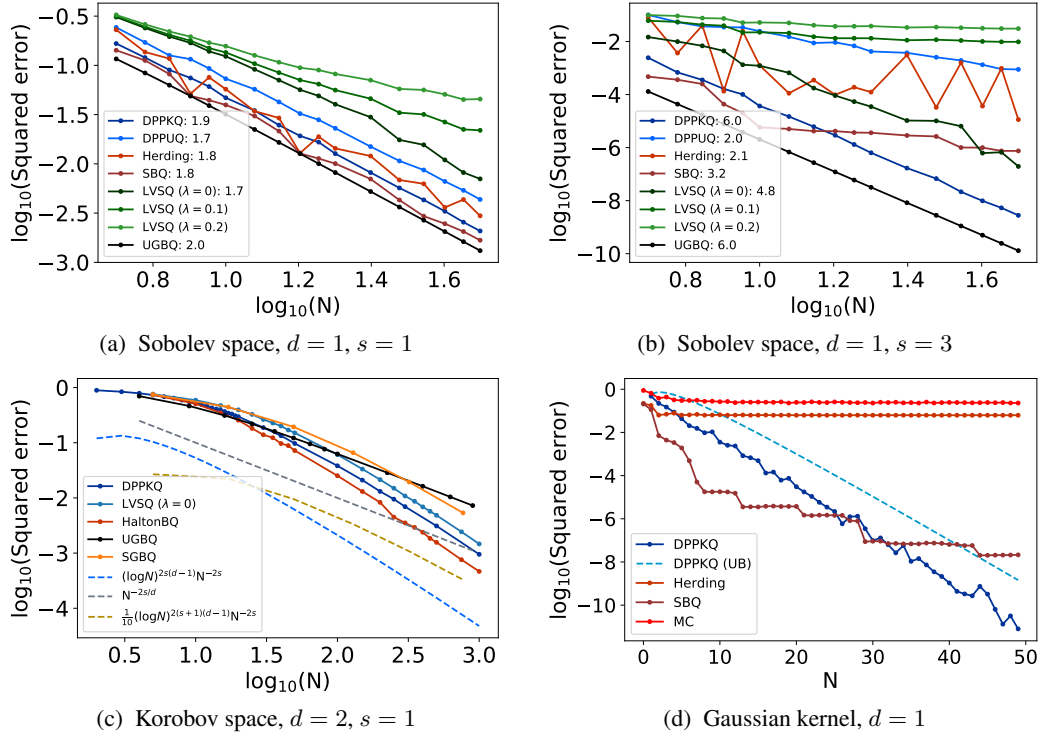

(a) Sobolev space, $d = 1$, $s = 1$

(b) Sobolev space, $d = 1$, $s = 3$

(c) Korobov space, $d = 2$, $s = 1$

(d) Gaussian kernel, $d = 1$

Figure 1: Squared error vs. number of nodes $N$ for various kernels.

and $\gamma = \frac{1}{2}$. Note that, this time, only the $y$-axis is on the log scale for better display, and that LVSQ is not plotted since we don't know how to sample from $q_\lambda$ in (6) in this case. We observe that the approximation error of DPPKQ converges to 0 as $\mathcal{O}(\alpha^N)$, while the discussion below Theorem 1 let us expect a slightly slower $\mathcal{O}(N\alpha^N)$. Herding improves slightly upon Monte Carlo that converges as $\mathcal{O}(N^{-1})$. Similarly to Sobolev spaces, the convergence of sequential Bayesian quadrature plateaus even if it has the smallest error for small $N$. We also conclude that DPPKQ is a close runner-up to SBQ and definitely takes the lead for large enough $N$.

## 6 Conclusion

In this article, we proposed a quadrature rule for functions living in a RKHS. The nodes are drawn from a DPP tailored to the RKHS kernel, while the weights are the solution to a tractable, non-regularized optimization problem. We proved that the expected value of the squared worst case error is bounded by a quantity that depends on the eigenvalues of the integral operator associated to the RKHS kernel, thus preserving the natural feel and the generality of the bounds for kernel quadrature [1]. Key intermediate quantities further have clear geometric interpretations in the ambient RKHS. Experimental comparisons suggest that DPP quadrature favourably compares with existing kernel-based quadratures. In specific cases where an optimal quadrature is known, such as the uniform grid for 1D periodic Sobolev spaces, DPPKQ seems to have the optimal convergence rate. However, our generic error bound does not reflect this optimality in the Sobolev case, and must thus be sharpened.

We have discussed room for improvement in our proofs. Further work should also address exact sampling algorithms, which do not exist yet when the spectral decomposition of the integral operator is not known. Approximate algorithms would also suffice, as long as the error bound is preserved.

### Acknowledgments

We acknowledge support from ANR grant BoB (ANR-16-CE23-0003) and région Hauts-de-France. We also thank Adrien Hardy and the reviewers for their detailed and insightful comments.

## Footnotes

[1]In the finite case, more common in ML, projection DPPs are also called *elementary* DPPs [26].

[2]The reconstruction operator $\boldsymbol{\Phi}$ depends on the nodes $x_j$, although our notation doesn't reflect it for simplicity.

[3]For $\mu \in \mathcal{F}, \boldsymbol{\Phi}^*\mu = (\mu(x_j))_{j \in [N]}$. $\boldsymbol{\Phi}^*\boldsymbol{\Phi}$ is an operator from $\mathbb{R}^N$ to $\mathbb{R}^N$ that can be identified with $\boldsymbol{K}(\boldsymbol{x})$.

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
