[Supplementary Material]

# Supplementary material for
# *Kernel quadrature with DPPs*

**Ayoub Belhadji,   Rémi Bardenet,   Pierre Chainais**
Univ. Lille, CNRS, Centrale Lille, UMR 9189 - CRIStAL, Villeneuve d'Ascq, France
{ayoub.belhadji, remi.bardenet, pierre.chainais}@univ-lille.fr

## A   Implementation details

In this section, we give details on the repulsion kernels in each example of the main paper, and explain how we sampled from the corresponding DPPs. In short, we relied on matrix models for univariate cases, and vanilla DPP sampling [10] for multivariate settings.

### A.1   The one-dimensional periodic Sobolev space

Consider the kernel $k_s : [0,1] \times [0,1] \to \mathbb{R}_+$ defined by

$$k_s(x,y) = 1 + \sum_{m \in \mathbb{N}^*} \frac{1}{m^{2s}} \cos(2\pi m(x-y)). \tag{1}$$

The Mercer decomposition of $k_s$ associated to the uniform measure $\mathrm{d}\omega$ on $[0,1]$ writes

$$k_s(x,y) = \sum_{k \in \mathbb{Z}} \frac{1}{\max(1,|k|)^{2s}} \, e^{2\pi i k x} e^{-2\pi i k y}. \tag{2}$$

The corresponding repulsion kernel is

$$\mathfrak{K}(x,y) = e^{\pi i N(x-y)} \sum_{m=-N/2}^{N/2} e^{2\pi i m x} e^{-2\pi i m y} = \sum_{m=0}^{N} e^{2\pi i m x} e^{-2\pi i m y}, \tag{3}$$

if $N$ is even and

$$\mathfrak{K}(x,y) = e^{\pi i (N-1)(x-y)} \sum_{m=-(N-1)/2}^{(N+1)/2} e^{2\pi i m x} e^{-2\pi i m y} = \sum_{m=0}^{N} e^{2\pi i m x} e^{-2\pi i m y}, \tag{4}$$

if not. The projection DPP with kernel $\mathfrak{K}$ and reference measure $\mathrm{d}\omega$ can be sampled through a matrix model. Indeed this DPP is also the distribution of the arguments (normalized by $2\pi$) of the eigenvalues of a random unitary matrix drawn from the Haar measure on $\mathbb{U}_{N+1}$ [25]. Sampling such matrices can be done, e.g., using the QR decomposition of a matrix with i.i.d. unit complex Gaussians as coefficients [16].

### A.2   The one-dimensional Gaussian kernel

Let $k_\gamma : \mathbb{R} \times \mathbb{R} \to \mathbb{R}_+$ and the reference measure $\mathrm{d}\omega$ be defined by

$$k_\gamma(x,y) = e^{-\frac{(x-y)^2}{2\gamma^2}}, \quad \mathrm{d}\omega(x) = \frac{1}{\sqrt{2\pi}\sigma} e^{-\frac{x^2}{2\sigma^2}}. \tag{5}$$

For notational convenience, we further let

$$a = \frac{1}{4\sigma^2}, \quad b = \frac{1}{2\gamma^2}, \quad c = \sqrt{a^2 + 2ab}, \tag{6}$$

and
$$A = a + b + c, \quad B = b/A. \tag{7}$$

Now, the Mercer decomposition of $k_\gamma$ reads [20]

$$k_\gamma(x, y) = \sum_{m \in \mathbb{N}} \sigma_m e_m(x) e_m(y), \tag{8}$$

where

$$\sigma_m = \sqrt{\frac{2a}{A}} B^m, \quad e_m = e^{-(c-a)x^2} H_m(\sqrt{2c}x), \tag{9}$$

and $H_m$ is the $m$-th Hermite polynomial (i.e., orthonormal polynomials for the pdf of a unit Gaussian). Now, denote

$$\tilde{e}_m(x) := H_m(\sqrt{2c}x), \tag{10}$$

and the measure

$$\mathrm{d}\tilde{\omega} = \frac{1}{\sqrt{2\pi}\sigma} e^{-2cx^2}. \tag{11}$$

The rescaled polynomials $(\tilde{e}_m)_{m \in \mathbb{N}}$ are orthonormal with respect to the measure $\mathrm{d}\tilde{\omega}$. Moreover, for $x \in \mathbb{R}$,

$$e_m(x)e_{m'}(x)\mathrm{d}\omega(x) = e^{-(c-a)x^2} H_m(\sqrt{2c}x) e^{-(c-a)x^2} H_{m'}(\sqrt{2c}x) \frac{1}{\sqrt{2\pi}\sigma} e^{-2ax^2} \tag{12}$$

$$= H_m(\sqrt{2c}x) H_{m'}(\sqrt{2c}x) e^{-2cx^2} \tag{13}$$

$$= \tilde{e}_m(x)\tilde{e}_{m'}(x)\mathrm{d}\tilde{\omega}(x). \tag{14}$$

Thus, for $\boldsymbol{x} = (x_i)_{i \in [N]} \in \mathbb{R}^N$, we have

$$\mathrm{Det}\,\boldsymbol{E}(\boldsymbol{x}) \otimes_{i \in [N]} \mathrm{d}\omega(x_i) = \mathrm{Det}\,\tilde{\boldsymbol{E}}(\boldsymbol{x}) \otimes_{i \in [N]} \mathrm{d}\tilde{\omega}(x_i). \tag{15}$$

In other words, the projection DPP associated to the orthonormal family $(e_n)_{n \in [N]}$ and the reference measure $\mathrm{d}\omega$ is equivalent to the projection DPP associated to the orthonormal family $(\tilde{e}_n)_{n \in [N]}$ and the reference measure $\mathrm{d}\tilde{\omega}$. The latter DPP is known to be the distribution of the eigenvalues of a symmetrized matrix with i.i.d. Gaussian entries [14], which is easily implemented.

## A.3 The case of a tensor product of RKHSs

We consider the case where $\mathcal{F}$ writes as a tensor product of RKHSs, with the associated kernel

$$k(\boldsymbol{x}, \boldsymbol{y}) = \prod_{\ell \in [L]} k_\ell(x_\ell, y_\ell), \tag{16}$$

with $k_\ell : \mathcal{X}_\ell \times \mathcal{X}_\ell \to \mathbb{R}$.

### A.3.1 The multivariate integral operator

The integral operator becomes

$$\boldsymbol{\Sigma} f(\boldsymbol{x}) = \int_{\mathcal{X}} f(\boldsymbol{x}) k(\boldsymbol{x}, \boldsymbol{y})\mathrm{d}\omega(\boldsymbol{y}) = \prod_{\ell \in [L]} \int_{\mathcal{X}_\ell} f_\ell(x_\ell) k_\ell(x_\ell, y_\ell)\mathrm{d}\omega_\ell(y_\ell). \tag{17}$$

In the main paper, we considered for instance the Korobov space $\mathbb{K}_s^d([0, 1])$, defined as the tensor product of unidimensional periodic Sobolev spaces. Note that an element $f$ of $\mathbb{K}_s^d([0, 1])$ is such that

$$\frac{\partial^{u_1 + \cdots + u_d}}{\partial x_1^{u_1} \ldots \partial x_d^{u_d}} f \in \mathbb{L}_2([0, 1]^d), \quad \forall u_1, \ldots u_d \in \{0, \ldots, s\}.$$

This implies that $\mathbb{K}_s^d([0, 1])$ is included in the multidimensional Sobolev space, which corresponds to the same requirement, but only for multi-indices such that $\|u_i\|_1 \leq s$. Another example, featured in this supplementary material, is the multidimensional Gaussian space associated to the Gaussian

kernel on $\mathcal{X}_\ell = \mathbb{R}$ and the multidimensional Gaussian measure. In this case, the kernel $k_{\gamma,d}$ can be written as the tensor product of the Gaussian kernels on $\mathbb{R}$:

$$\forall \boldsymbol{x}, \boldsymbol{y} \in \mathbb{R}^d, \ k_{\gamma,d}(\boldsymbol{x}, \boldsymbol{y}) = \prod_{i \in [d]} k_\gamma(x_i, y_i). \tag{18}$$

In general, the eigenpairs of the integral operator are the tensor products of the eigenpairs of the integral operators $\Sigma_\ell$ corresponding to the spaces $\mathcal{F}_\ell$ and measures $\mathrm{d}\omega_\ell$. In other words, for $\boldsymbol{u} \in (\mathbb{N} \setminus \{0\})^d$,

$$\boldsymbol{\Sigma} \otimes_{i \in [d]} e_{\ell, u_i} = \prod_{i \in [d]} \sigma_{\ell, u_i} \otimes_{i \in [d]} e_{\ell, u_i}. \tag{19}$$

### A.3.2 Fixing an order on multi-indices

The definition of the projection DPP and its kernel $\mathfrak{K}$ now require that we fix an order on multi-indices. We choose an order $\prec$ that keeps eigenvalues decreasing, as in the univariate case where $\sigma_1 \geq \sigma_2 \geq \dots$. Whenever the univariate eigenvalues take the form $\sigma_i = \frac{1}{(1+i)^\eta}$ with $\eta > 0$, such as in the Korobov case, it holds

$$\prod_{i \in [d]} \sigma_{u_i} \leq \prod_{i \in [d]} \sigma_{v_i} \Leftrightarrow \left( \prod_{i \in [d]} \frac{1}{1+u_i} \right)^\eta \leq \left( \prod_{i \in [d]} \frac{1}{1+v_i} \right)^\eta \tag{20}$$

$$\Leftrightarrow \sum_{i \in [d]} \log(1 + v_i) \leq \sum_{i \in [d]} \log(1 + u_i). \tag{21}$$

Now, if the eigenvalues takes the form $\sigma_i = \eta^{-i}$, with $\eta > 1$, as in the Gaussian case,

$$\prod_{i \in [d]} \sigma_{u_i} \leq \prod_{i \in [d]} \sigma_{v_i} \Leftrightarrow \prod_{i \in [d]} \frac{1}{\eta^{u_i}} \leq \prod_{i \in [d]} \frac{1}{\eta^{v_i}} \tag{22}$$

$$\Leftrightarrow \left( \sum_{i \in [d]} v_i \right) \log \eta \leq \left( \sum_{i \in [d]} u_i \right) \log \eta \tag{23}$$

$$\Leftrightarrow \sum_{i \in [d]} v_i \leq \sum_{i \in [d]} u_i. \tag{24}$$

In the multivariate Korobov and the Gaussian cases, we thus define in this work $\boldsymbol{u} \prec \boldsymbol{v}$ as (21) or (24), respectively.

Now, for $N \in \mathbb{N}$, let $\mathbf{u}_N = (\mathbf{u}_{1,N}, \dots, \mathbf{u}_{d,N}) \in \mathbb{N}^d$ be the $N$-th multi-index according to $\prec$. The repulsion kernel is defined as

$$\mathfrak{K}(x, y) = \sum_{n \in [N]} \prod_{i \in [d]} e_{\mathbf{u}_{i,N}}(x_i) e_{\mathbf{u}_{i,N}}(y_i), \quad x, y \in \mathbb{R}^d. \tag{25}$$

We sampled from the corresponding DPP using the generic sampling algorithm in [10], using the uniform and Gaussian distributions as proposal in the successive rejection sampling steps for the Korobov and Gaussian cases, respectively.

## B Supplementary simulations

In this section, we give more plots of the convergence of the quadrature error. Before that, we experimentally assess whether the upper bounds given in [1] are sharp. The author proved upper bounds for $\sigma_{N+1}$ in cases where the univariate eigenvalues $\sigma_{\ell,N}$ decrease polynomially or geometrically in $N$. In particular, for the Korobov spaces of dimension $d$ and regularity $s$, we have

$$\sigma_{N+1} = \mathcal{O}\left( (\log N)^{2s(d-1)} N^{-2s} \right). \tag{26}$$

For the Gaussian RKHS in dimension $d$, it holds

$$\sigma_{N+1} = \mathcal{O}\left( \beta^d e^{-\delta d!^{1/d} N^{1/d}} \right), \tag{27}$$

Figure 1: (Left): comparison of $\sigma_{N+1}$ in the Korobov case according to the spectral order and $(\log N)^{2s(d-1)}N^{-2s}$ for $d \in \{2, 3, 4\}$ and $s = 1$, (Right): comparison of $\sigma_{N+1}$ in the Gaussian case according to the spectral order and $\beta^d e^{-\delta d!^{1/d}N^{1/d}}$ for $d \in \{2, 3, 4\}$ and $\gamma = 1$.

where $\beta \in ]0, 1[$ and $\delta > 0$ are constants depending on the scale parameters of the kernel and the measure $d\omega$. In our experiments, we compare the errors of various quadratures to the two rates (26) and (27). We mean these rates to be proxies for plotting $\sigma_{\mathbf{u}_N}$, where $\mathbf{u}_N$ refers to the order introduced in Section A.3. Figure 1 shows that in the Korobov case, the rate (26) is indeed close to the corresponding eigenvalue for large values of $N$. The value of $(\log N)^{2s(d-1)}N^{-2s}$ could be larger than 1 for $d \geq 4$ and small values of $N$. As for the Gaussian case, Figure 1 shows that the rate (27) is also close to the corresponding eigenvalue for all values of $N$.

## B.1    The multi Fourier ensemble and Korobov RKHS

We consider the case of Korobov spaces with $d \in \{2, 3\}$ and $s \in \{1, 2\}$ and compare the quadrature error of the same algorithms as in 5.1. The results are compiled in Figure 2. The numerical simulations confirm the dependencies of the theoretical bounds of the different algorithms to the dimension $d$ and the regularity $s$. In particular, UGBQ have better performance for high values of $s$ and low values of $N$ while its asymptotic behaviour is still the same $\mathcal{O}(N^{-2s/d})$. Moreover, the empirical rate of SGBQ is similar to its theoretical rate $\mathcal{O}((\log N)^{2(s+1)(d-1)}N^{-2s})$ [9, 21]. Finally, the rate $\mathcal{O}((\log N)^{2s(d-1)}N^{-2s})$ is confirmed also for the algorithms DPPKQ, LVSQ $(\lambda = 0)$ and HaltonBQ.

Figure 2: The squared error for the Korobov space ($d \in \{2, 3\}$, $s \in \{1, 2\}$): (Top) the results for the dimension $d = 2$, (Left) the results for the regularity $s = 1$.

Figure 3: The squared error for the Gaussian space ($d \in \{2, 3\}$, $\gamma = 1$).

## B.2 The multi Gaussian ensemble

We consider the case of Gaussian spaces with $d \in \{2, 3\}$. The kernel $k_{\gamma,d}$ and the reference measure are the tensor product of respectively the same kernel and the same measure used in Section 5.2. We compare DPPKQ and Bayesian quadrature based on the tensor product of Gauss-Hermite nodes noted GHBQ. Note that a variant of this algorithm was proposed in [12]: the quadrature nodes are the tensor product of the Gauss-Hermite nodes however the weights were calculated differently. The authors proved under an assumption on the stability of the weights (that was verified empirically) that the rate of convergence is $\mathcal{O}(dr^d \beta'^d e^{-\delta' d N^{1/d}})$, where $r$ is a constant that quantify the stability of the weights, and $\beta'$, $\delta'$ are constants that depend simultaneously on the the stability of the weights and length scales of the kernel and the measure. The results are compiled in Figure 3.

The numerical simulations shows that the empirical rate of DPPKQ is $\mathcal{O}(e^{-\delta d N^{1/d}})$ that is slightly better than its theoretical rate $\mathcal{O}(e^{-\delta d!^{1/d} N^{1/d}})$. Moreover, we observe that the empirical rate of DPPKQ is better than the empirical rate of HGBQ.

# C  Mercer's theorem, leverage scores, and principal angles

For the sake of completeness, this section gathers some known results, which will be used to prove our own. We will need a general version of Mercer's theorem, as usual for kernel methods, see Section C.1. On a more technical ground, we will also need formulas for leverage score changes under rank 1 updates, see Section C.2. Finally, Section C.3 covers principal angles between subspaces of a Hilbert space, which bridge the gap between pairs of Hilbert subspaces and determinants, and facilitate taking expectations in Theorem 1.

## C.1  Mercer decomposition in non-compact subspaces

In this section we recall Mercer's theorem and its extensions to non-compact spaces. Let $\mathcal{X}$ be a measurable space and $\mathrm{d}\omega$ a measure on $\mathcal{X}$. Assume $k$ is a positive definite kernel on $\mathcal{X}$. Whenever it is well-defined, we consider the operator

$$\boldsymbol{\Sigma} f(x) = \int_{\mathcal{X}} k(x,y)f(y)\mathrm{d}\omega(y). \tag{28}$$

**Theorem 2.** *Assume that $\mathcal{X}$ is a compact space and $\mathrm{d}\omega$ is a finite Borel measure on $\mathcal{X}$. Then, there exists an orthonormal basis $(e_n)_{n\in\mathbb{N}^*}$ of $\mathbb{L}_2(\mathrm{d}\omega)$ consisting of eigenfunctions of $\boldsymbol{\Sigma}$, and the corresponding eigenvalues are non-negative. The eigenfunctions corresponding to non-vanishing eigenvalues can be taken to be continuous, and the kernel $k$ writes*

$$k(x,y) = \sum_{n\in\mathbb{N}^*} \sigma_n e_n(x)e_n(y), \tag{29}$$

*where the convergence is absolute and uniform.*

Theorem 2 was first proven when $\mathcal{X} = [0,1]$ and $\mathrm{d}\omega$ is the Lebesgue measure in [15]. A modern proof can be found in [13], while the proof in the general case can be found in [4]. Note, however, that the compactness assumption in Theorem 2 excludes kernels such as the Gaussian or the Laplace kernels. Hence, extensions to non-compact spaces are usually required in ML. In [24], the author extended Theorem 2 to $X = \cup_{i\in\mathbb{N}} X_i$, with the $X_i$s compact and $\mathrm{d}\omega(X_i) < \infty$. One can also extend Mercer's theorem under a *compact embedding* assumption [23]: the RKHS $\mathcal{F}$ associated to $k$ is said to be compactly embedded in $\mathbb{L}_2(\mathrm{d}\omega)$ if the application

$$I_{\mathcal{F}} : \mathcal{F} \longrightarrow \mathbb{L}_2(\mathrm{d}\omega)$$
$$f \longmapsto f$$

is compact. A sufficient condition for this assumption is the integrability of the diagonal (Lemma 2.3, [23]):

$$\int_{\mathcal{X}} k(x,x)\mathrm{d}\omega(x) < \infty. \tag{30}$$

Note that this condition is not necessary (Example 2.9, [23]). Now, under the compact embedding assumption, the pointwise convergence of the Mercer decomposition to the kernel $k$ is equivalent to the injectivity of the embedding $I_{\mathcal{F}}$ (Theorem 3.1, [23]).

## C.2  Leverage score changes under rank 1 updates

In this section we prove a lemma inspired from Lemma 5 in [3]. This lemma concerns the changes of leverage scores under rank 1 updates.

We start by recalling the definition of leverage scores, which play an important role in randomized linear algebra [6]. Let $N, M \in \mathbb{N}^*$, $M \geq N$. Let $\boldsymbol{A} \in \mathbb{R}^{N\times M}$ be a matrix of full rank. For $i \in [M]$, denote $\boldsymbol{a}_i$ the $i$-th column of the matrix $\boldsymbol{A}$. Now, the $i$-th leverage score of the matrix $\boldsymbol{A}$ is defined by

$$\tau_i(\boldsymbol{A}) = \boldsymbol{a}_i^{\mathsf{T}}(\boldsymbol{A}\boldsymbol{A}^{\mathsf{T}})^{-1}\boldsymbol{a}_i, \tag{31}$$

while the cross-leverage score between the $i$-th column and the $j$-th column is defined by

$$\tau_{i,j}(\boldsymbol{A}) = \boldsymbol{a}_i^{\mathsf{T}}(\boldsymbol{A}\boldsymbol{A}^{\mathsf{T}})^{-1}\boldsymbol{a}_j. \tag{32}$$

It holds [6]

$$\forall i \in [M],\ \tau_i(\boldsymbol{A}) \in [0,1], \tag{33}$$

and we have the following result.

**Lemma 3.** *Let $N, M \in \mathbb{N}^*$, $M \geq N$. Let $\boldsymbol{A} \in \mathbb{R}^{N \times M}$ of full rank and $\rho \in \mathbb{R}_+^*$ and $i \in [M]$. Let $\boldsymbol{W} \in \mathbb{R}^{M \times M}$ a diagonal matrix such that $\boldsymbol{W}_{i,i} = \sqrt{1+\rho}$ and $\boldsymbol{W}_{j,j} = 1$ for $j \neq i$. Then*

$$\tau_i(\boldsymbol{AW}) = \frac{(1+\rho)\tau_i(\boldsymbol{A})}{1 + \rho\tau_i(\boldsymbol{A})} \geq \tau_i(\boldsymbol{A}), \tag{34}$$

*and*

$$\forall j \in [M] - \{i\}, \ \tau_j(\boldsymbol{AW}) = \tau_j(\boldsymbol{A}) - \frac{\rho\tau_{i,j}(\boldsymbol{A})^2}{1 + \rho\tau_i(\boldsymbol{A})} \leq \tau_j(\boldsymbol{A}). \tag{35}$$

The proof of this lemma is similar to Lemma 5 in [3]. We recall the proof for completeness.

*Proof.* (Adapted from [3]) The Sherman-Morrison formula applied to $\boldsymbol{AWW}^\intercal\boldsymbol{A}^\intercal$ and the vector $\sqrt{\rho}\boldsymbol{a}_i$ yields

$$(\boldsymbol{AWW}^\intercal\boldsymbol{A}^\intercal)^{-1} = (\boldsymbol{AA}^\intercal + \rho\boldsymbol{a}_i\boldsymbol{a}_i^\intercal)^{-1} \tag{36}$$

$$= (\boldsymbol{AA}^\intercal)^{-1} - \frac{(\boldsymbol{AA}^\intercal)^{-1}\rho\boldsymbol{a}_i\boldsymbol{a}_i^\intercal(\boldsymbol{AA}^\intercal)^{-1}}{1 + \rho\boldsymbol{a}_i^\intercal(\boldsymbol{AA}^\intercal)^{-1}\boldsymbol{a}_i}. \tag{37}$$

By definition of $\tau_i(\boldsymbol{AW})$

$$\tau_i(\boldsymbol{AW}) = \sqrt{1+\rho}\,\boldsymbol{a}_i^\intercal(\boldsymbol{AWW}^\intercal\boldsymbol{A}^\intercal)^{-1}\boldsymbol{a}_i\sqrt{1+\rho} \tag{38}$$

$$= (1+\rho)\boldsymbol{a}_i^\intercal\left((\boldsymbol{AA}^\intercal)^{-1} - \frac{(\boldsymbol{AA}^\intercal)^{-1}\rho\boldsymbol{a}_i\boldsymbol{a}_i^\intercal(\boldsymbol{AA}^\intercal)^{-1}}{1 + \rho\boldsymbol{a}_i^\intercal(\boldsymbol{AA}^\intercal)^{-1}\boldsymbol{a}_i}\right)\boldsymbol{a}_i$$

$$= (1+\rho)\left(\tau_i(\boldsymbol{A}) - \frac{\rho\tau_i(\boldsymbol{A})^2}{1 + \rho\tau_i(\boldsymbol{A})}\right)$$

$$= (1+\rho)\frac{\tau_i(\boldsymbol{A})}{1 + \rho\tau_i(\boldsymbol{A})}.$$

Now let $j \in [M] - \{i\}$. By definition of $\tau_j(\boldsymbol{AW})$

$$\tau_j(\boldsymbol{AW}) = \boldsymbol{a}_j^\intercal(\boldsymbol{AWW}^\intercal\boldsymbol{A}^\intercal)^{-1}\boldsymbol{a}_j \tag{39}$$

$$= \boldsymbol{a}_j^\intercal\left((\boldsymbol{AA}^\intercal)^{-1} - \frac{(\boldsymbol{AA}^\intercal)^{-1}\rho\boldsymbol{a}_i\boldsymbol{a}_i^\intercal(\boldsymbol{AA}^\intercal)^{-1}}{1 + \rho\boldsymbol{a}_i^\intercal(\boldsymbol{AA}^\intercal)^{-1}\boldsymbol{a}_i}\right)\boldsymbol{a}_j$$

$$= \tau_j(\boldsymbol{A}) - \frac{\rho\tau_{i,j}(\boldsymbol{A})^2}{1 + \rho\tau_i(\boldsymbol{A})}$$

$$\leq \tau_j(\boldsymbol{A}).$$

$\square$

### C.3 Principal angles between subspaces in Hilbert spaces

We recall in this section the definition of principal angles between subspaces in Hilbert spaces and connect them to the determinant of the Gramian matrix of their orthonormal bases.

**Proposition 5.** *Let $\mathcal{H}$ be a Hilbert space. Let $\mathcal{P}_1$ and $\mathcal{P}_2$ be two finite-dimensional subspaces of $\mathcal{H}$ with $N = \dim \mathcal{P}_1 = \dim \mathcal{P}_2$. Denote $\boldsymbol{\Pi}_{\mathcal{P}_1}$ and $\boldsymbol{\Pi}_{\mathcal{P}_2}$ the orthogonal projections of $\mathcal{H}$ onto these two subspaces. There exist two orthonormal bases for $\mathcal{P}_1$ and $\mathcal{P}_2$ denoted $(\boldsymbol{v}_i^1)_{i \in [N]}$ and $(\boldsymbol{v}_i^2)_{i \in [N]}$, and a set of angles $\theta_i(\mathcal{P}_1, \mathcal{P}_2) \in [0, \frac{\pi}{2}]$ such that*

$$\cos\theta_N(\mathcal{P}_1, \mathcal{P}_2) \leq \cdots \leq \cos\theta_1(\mathcal{P}_1, \mathcal{P}_2), \tag{40}$$

*and for $i \in [1, ..., N]$*

$$\langle \boldsymbol{v}_i^1, \boldsymbol{v}_i^2 \rangle_\mathcal{H} = \cos\theta_i(\mathcal{P}_1, \mathcal{P}_2), \tag{41}$$

*and*

$$\boldsymbol{\Pi}_{\mathcal{P}_1}\boldsymbol{v}_i^2 = \cos\theta_i(\mathcal{P}_1, \mathcal{P}_2)\boldsymbol{v}_i^1, \tag{42}$$

*and*

$$\boldsymbol{\Pi}_{\mathcal{P}_2}\boldsymbol{v}_i^1 = \cos\theta_i(\mathcal{P}_1, \mathcal{P}_2)\boldsymbol{v}_i^2. \tag{43}$$

*In particular*

$$\cos\theta_N(\mathcal{P}_1, \mathcal{P}_2) = \inf_{\boldsymbol{v} \in \mathcal{P}_1, \|\boldsymbol{v}\|_\mathcal{H} = 1} \|\boldsymbol{\Pi}_{\mathcal{P}_2}\boldsymbol{v}\|_\mathcal{H}. \tag{44}$$

We refer to [8] for the proof in the finite-dimensional case and [5] for the general case. The following result shows that the principal angles are somewhat independent of the choice of orthonormal bases. It can be found in [2, 17] for the finite dimensional case. We give here the proof for the general case, for the sake of completeness.

**Corollary 2.** *Let $(w_i^1)_{i\in[N]}$ be any orthonormal basis of $\mathcal{P}_1$ and $(w_i^2)_{i\in[N]}$ be any orthonormal basis of $\mathcal{P}_2$, and let $W = (\langle w_i^1, w_j^2\rangle_{\mathcal{H}})_{1\leq i,j\leq N}$ and $G = WW^\intercal$. Then the eigenvalues of $G$ are the $\cos^2\theta_i(\mathcal{P}_1,\mathcal{P}_2)$. In particular, $\mathrm{Det}^2 W = \mathrm{Det}\, G = \prod_{i\in[N]} \cos^2\theta_i(\mathcal{P}_1,\mathcal{P}_2)$.*

*Proof.* Let $(v_i^i)_{i\in[N]}$, $i \in \{1,2\}$, be the bases of Proposition 5. Let $U^1 \in \mathbb{O}_N(\mathbb{R})$ be such that

$$\forall i \in [N],\ w_i^1 = \sum_{j\in[N]} u_{i,j}^1 v_j^1. \tag{45}$$

Similarly, there exists a matrix $U^2 \in \mathbb{O}_N(\mathbb{R})$ such that

$$\forall i \in [N],\ w_i^2 = \sum_{j\in[N]} u_{i,j}^2 v_j^2. \tag{46}$$

This implies that

$$W = U^1 V U^{2\,\intercal}, \tag{47}$$

where $V = (\langle v_i^1, v_j^2\rangle_{\mathcal{H}})_{1\leq i,j\leq N}$. Then

$$G = WW^\intercal = U^1 V V^\intercal U^{1\,\intercal}. \tag{48}$$

Thus the eigenvalues of $G$ are the eigenvalues of $VV^\intercal$. By Proposition 5, the diagonal elements of $V$ are

$$v_{i,i} = \langle v_i^1, v_i^2\rangle_{\mathcal{H}} = \cos\theta_i(\mathcal{P}_1,\mathcal{P}_2). \tag{49}$$

We finish the proof by showing that the anti-diagonal elements satisfy

$$v_{i,j} = \langle v_i^1, v_j^2\rangle_{\mathcal{H}} = 0. \tag{50}$$

By (42),

$$\forall i \in [N],\ \sum_{j\in[N]} \langle v_i^2, v_j^1\rangle_{\mathcal{H}}^2 = \|\Pi_{\mathcal{P}_1} v_i^2\|_{\mathcal{H}}^2 = \cos^2\theta_i(\mathcal{P}_1,\mathcal{P}_2). \tag{51}$$

Then

$$\sum_{i\in[N]}\sum_{j\in[N]} \langle v_i^2, v_j^1\rangle_{\mathcal{H}}^2 = \sum_{i\in[N]} \cos^2\theta_i(\mathcal{P}_1,\mathcal{P}_2) = \sum_{i\in[N]} \langle v_i^2, v_i^1\rangle_{\mathcal{H}}^2. \tag{52}$$

Thus

$$\sum_{\substack{i,j\in[N]\\ i\neq j}} \langle v_i^2, v_i^1\rangle_{\mathcal{H}}^2 = 0. \tag{53}$$

Finally, $V$ is a diagonal matrix and the eigenvalues of $G$ are the $\cos^2\theta_i(\mathcal{P}_1,\mathcal{P}_2)$. $\qquad\square$

# D  Proofs of our results

Section D.1 contains the proof of Proposition 2. In the main paper, we use it under the form of Corollary 1 to ensure that $K(x)$ is almost surely invertible when $x = \{x_1,\ldots,x_N\}$ is a projection DPP with reference measure $d\omega$ and kernel (10). This allows computing the quadrature weights.

The rest of Section D deals with Theorem 1, our upper bound on the approximation error of DPP-based kernel quadrature. The proof is rather long, but can be decomposed in four steps, which we now introduce for ease of reading.

First, we prove Lemma 1, which separates the search for an upper bound into examining the contribution of the three terms in (17); this is Section D.2. The first two terms of (17) only depend on the function $g$ in (1), and we leave them be. The third term is more geometric, and relates to the approximation error of the space spanned by $(e_n^{\mathcal{F}})_{n\in[N]}$ by the (random) subspace $\mathcal{T}(x)$.

Second, in Section D.3, we bound this geometric term for a fixed DPP realization $\boldsymbol{x}$. We pay attention to obtain a bound that will later yield a tractable expectation under that DPP. This is done in Proposition 4, which in turn requires two intermediate results, Lemma 4 and Proposition 6.

Third, we take the expectation of the bound in Proposition 4 under the proposed DPP. This is done in Proposition 3, which is proven thanks to Proposition 2, Lemmas 2, 5 & 6. This is Section D.4.

Fourth, Theorem 1 is obtained in Section D.5, using the results of the previous steps, and an argument to reduce the proof to RKHSs with flat initial spectrum.

## D.1 Proof of Proposition 2

*Proof.* Recall the Mercer decomposition of $k$:

$$k(x, y) = \sum_{m \in \mathbb{N}^*} \sigma_m e_m(x) e_m(y), \tag{54}$$

where the convergence is point-wise on $\mathcal{X}$. Define for $M \in \mathbb{N}^*$, $M \geq N$ the $M$-th truncated kernel

$$k_M(x, y) = \sum_{m \in [M]} \sigma_m e_m(x) e_m(y). \tag{55}$$

By (54)

$$\forall x, y \in \mathcal{X}, \ \lim_{M \to \infty} k_M(x, y) = k(x, y). \tag{56}$$

Let $\boldsymbol{x} = (x_1, \ldots, x_N) \in \mathcal{X}^N$ such that $\mathrm{Det}\, \boldsymbol{E}(\boldsymbol{x}) \neq 0$, and define

$$\boldsymbol{K}_M(\boldsymbol{x}) = (k_M(x_i, x_j))_{i,j \in [N]}. \tag{57}$$

By the continuity of the function $\boldsymbol{M} \in \mathbb{R}^{N \times N} \mapsto \mathrm{Det}\, \boldsymbol{M}$ and by (56)

$$\lim_{M \to \infty} \mathrm{Det}\, \boldsymbol{K}_M(\boldsymbol{x}) = \mathrm{Det}\, \boldsymbol{K}(\boldsymbol{x}). \tag{58}$$

Thus to prove that $\mathrm{Det}\, \boldsymbol{K}(\boldsymbol{x}) > 0$, it is enough to prove that the $\mathrm{Det}\, \boldsymbol{K}_M(\boldsymbol{x})$ is larger than a positive real number for $M$ large enough. We write

$$\boldsymbol{K}_M(\boldsymbol{x}) = \boldsymbol{F}_M(\boldsymbol{x})^\mathsf{T} \boldsymbol{\Sigma}_M \boldsymbol{F}_M(\boldsymbol{x}), \tag{59}$$

with $\boldsymbol{F}_M(\boldsymbol{x}) = (e_i(x_j))_{(i,j) \in [M] \times [N]}$ and $\Sigma_M$ is a diagonal matrix containing the first $M$ eigenvalues $(\sigma_m)$. The Cauchy-Binet identity yields

$$\mathrm{Det}\, \boldsymbol{K}_M(\boldsymbol{x}) = \sum_{T \subset [M], |T| = N} \prod_{i \in T} \sigma_i \, \mathrm{Det}^2(e_i(x_j))_{(i,j) \in T \times [N]} \tag{60}$$

$$\geq \prod_{i \in [N]} \sigma_i \, \mathrm{Det}^2 \boldsymbol{E}(\boldsymbol{x}) > 0. \tag{61}$$

Therefore,

$$\mathrm{Det}\, \boldsymbol{K}(\boldsymbol{x}) = \lim_{M \to \infty} \mathrm{Det}\, \boldsymbol{K}_M(\boldsymbol{x}) \geq \prod_{i \in [N]} \sigma_i \, \mathrm{Det}^2 \boldsymbol{E}(\boldsymbol{x}) > 0. \tag{62}$$

so that $\boldsymbol{K}(\boldsymbol{x})$ is a.s. invertible. $\qquad \square$

## D.2 Proof of Lemma 1

*Proof.* First, we prove that

$$\|\boldsymbol{\Sigma}^{-1/2} \mu_g\|_{\mathcal{F}}^2 = \|g\|_{\mathrm{d}\omega}^2 \leq 1. \tag{63}$$

Recall that

$$\mu_g = \int_{\mathcal{X}} g(y) k(., y) \mathrm{d}\omega(y), \tag{64}$$

and that we assumed in Section 1 that $\mathcal{F}$ is dense in $\mathbb{L}_2(\mathrm{d}\omega)$, so that $(e_m)_{m \in \mathbb{N}}$ is an orthonormal basis of $\mathbb{L}_2(\mathrm{d}\omega)$ and the eigenvalues $\sigma_n$ are strictly positive. Now let $\boldsymbol{\Sigma}^{-1/2} : \mathcal{F} \to \mathbb{L}_2(\mathrm{d}\omega)$ and $\boldsymbol{\Sigma}^{1/2} : \mathbb{L}_2(\mathrm{d}\omega) \to \mathcal{F}$ be defined by

$$\boldsymbol{\Sigma}^{-1/2} e_m^{\mathcal{F}} = e_m, \ \forall m \in \mathbb{N}^*, \tag{65}$$

$$\mathbf{\Sigma}^{1/2}e_m = e_m^{\mathcal{F}}, \ \forall m \in \mathbb{N}^*. \tag{66}$$

Observe that $\mathbf{\Sigma}^{-1/2}\mu_g = \mathbf{\Sigma}^{-1/2}\mathbf{\Sigma}g = \mathbf{\Sigma}^{1/2}g \in \mathcal{F}$. Now, for $m \in \mathbb{N}^*$,

$$\langle e_m^{\mathcal{F}}, \mathbf{\Sigma}^{-1/2}\mu_g \rangle_{\mathcal{F}} = \langle e_m^{\mathcal{F}}, \mathbf{\Sigma}^{1/2}g \rangle_{\mathcal{F}} \tag{67}$$

$$= \langle e_m^{\mathcal{F}}, \mathbf{\Sigma}^{1/2} \sum_{n \in \mathbb{N}^*} \langle g, e_n \rangle_{\mathrm{d}\omega} e_n \rangle_{\mathcal{F}} \tag{68}$$

$$= \sum_{n \in \mathbb{N}^*} \langle g, e_n \rangle_{\mathrm{d}\omega} \langle e_m^{\mathcal{F}}, \mathbf{\Sigma}^{1/2}e_n \rangle_{\mathcal{F}} \tag{69}$$

$$= \sum_{n \in \mathbb{N}^*} \langle g, e_n \rangle_{\mathrm{d}\omega} \langle e_m^{\mathcal{F}}, e_n^{\mathcal{F}} \rangle_{\mathcal{F}} \tag{70}$$

$$= \langle g, e_m \rangle_{\mathrm{d}\omega}.$$

As a consequence,

$$\|\mathbf{\Sigma}^{-1/2}\mu_g\|_{\mathcal{F}}^2 = \|g\|_{\mathrm{d}\omega}^2 \le 1. \tag{71}$$

Now we turn to proving (17) from the main text. Define first the operators $\mathbf{\Sigma}_N, \mathbf{\Sigma}_N^{1/2}, \mathbf{\Sigma}_N^{\perp}, \mathbf{\Sigma}_N^{\perp 1/2}$ : $\mathbb{L}_2(\mathrm{d}\omega) \to \mathcal{F}, \mathbf{\Sigma}_N^{1/2} : \mathbb{L}_2(\mathrm{d}\omega) \to \mathcal{F}$ and $\mathbf{\Sigma}_N^{\perp} : \mathbb{L}_2(\mathrm{d}\omega) \to \mathcal{F}$ by

$$\mathbf{\Sigma}_N e_m = \begin{cases} \sigma_m e_m & \text{if } m \in [N] \\ 0 & \text{else} \end{cases}, \tag{72}$$

$$\mathbf{\Sigma}_N^{1/2} e_m = \begin{cases} \sqrt{\sigma_m} e_m & \text{if } m \in [N] \\ 0 & \text{else} \end{cases}, \tag{73}$$

$$\mathbf{\Sigma}_N^{\perp} e_m = \begin{cases} 0 & \text{if } m \in [N] \\ \sigma_m e_m & \text{if } m \ge N+1 \end{cases}, \tag{74}$$

$$\mathbf{\Sigma}_N^{\perp 1/2} e_m = \begin{cases} 0 & \text{if } m \in [N] \\ \sqrt{\sigma_m} e_m & \text{if } m \ge N+1 \end{cases}, \tag{75}$$

Note that $\mathbf{\Sigma}^{1/2} = \mathbf{\Sigma}_N^{1/2} + \mathbf{\Sigma}_N^{\perp 1/2}$ and

$$\sup_{\|\mu\|_{\mathcal{F}} \le 1} \|\mathbf{\Sigma}_N^{\perp 1/2}\mu\|_{\mathcal{F}}^2 = \sigma_{N+1}. \tag{76}$$

Using (71), there exists $\tilde{\mu}_g \in \mathcal{F}$ such that $\|\tilde{\mu}_g\|_{\mathcal{F}} \le 1$ and $\mu_g = \mathbf{\Sigma}^{1/2}\tilde{\mu}_g$. Now, the approximation error writes

$$\|\mathbf{\Pi}_{\mathcal{T}(\boldsymbol{x})^{\perp}}\mu_g\|_{\mathcal{F}}^2 = \|\mathbf{\Pi}_{\mathcal{T}(\boldsymbol{x})^{\perp}}\mathbf{\Sigma}^{1/2}\tilde{\mu}_g\|_{\mathcal{F}}^2 \tag{77}$$

$$= \|\mathbf{\Pi}_{\mathcal{T}(\boldsymbol{x})^{\perp}}(\mathbf{\Sigma}_N^{1/2} + \mathbf{\Sigma}_N^{\perp 1/2})\tilde{\mu}_g\|_{\mathcal{F}}^2$$

$$= \|\mathbf{\Pi}_{\mathcal{T}(\boldsymbol{x})^{\perp}}\mathbf{\Sigma}_N^{1/2}\tilde{\mu}_g\|_{\mathcal{F}}^2 + \|\mathbf{\Pi}_{\mathcal{T}(\boldsymbol{x})^{\perp}}\mathbf{\Sigma}_N^{\perp 1/2}\tilde{\mu}_g\|_{\mathcal{F}}^2 \tag{78}$$

$$+ 2\langle \mathbf{\Pi}_{\mathcal{T}(\boldsymbol{x})^{\perp}}\mathbf{\Sigma}_N^{1/2}\tilde{\mu}_g, \mathbf{\Pi}_{\mathcal{T}(\boldsymbol{x})^{\perp}}\mathbf{\Sigma}_N^{\perp 1/2}\tilde{\mu}_g \rangle_{\mathcal{F}}$$

$$\le 2\left(\|\mathbf{\Pi}_{\mathcal{T}(\boldsymbol{x})^{\perp}}\mathbf{\Sigma}_N^{1/2}\tilde{\mu}_g\|_{\mathcal{F}}^2 + \|\mathbf{\Pi}_{\mathcal{T}(\boldsymbol{x})^{\perp}}\mathbf{\Sigma}_N^{\perp 1/2}\tilde{\mu}_g\|_{\mathcal{F}}^2\right).$$

The operator $\mathbf{\Pi}_{\mathcal{T}(\boldsymbol{x})^{\perp}}$ is an orthogonal projection and $\|\tilde{\mu}_g\|_{\mathcal{F}} \le 1$ so that by (76)

$$\|\mathbf{\Pi}_{\mathcal{T}(\boldsymbol{x})^{\perp}}\mathbf{\Sigma}_N^{\perp 1/2}\tilde{\mu}_g\|_{\mathcal{F}}^2 \le \|\mathbf{\Sigma}_N^{\perp 1/2}\tilde{\mu}_g\|_{\mathcal{F}}^2 \le \sigma_{N+1}. \tag{79}$$

Now, recall that the $(e_n^{\mathcal{F}})_{n \in [N]}$ is orthonormal. Moreover for $n \in [N]$, $e_n^{\mathcal{F}}$ is an eigenfunction of $\mathbf{\Sigma}_N^{1/2}$ and the corresponding eigenvalue is $\sqrt{\sigma}_n$. Thus

$$\mathbf{\Pi}_{\mathcal{T}(\boldsymbol{x})^{\perp}}\mathbf{\Sigma}_N^{1/2}\tilde{\mu}_g = \mathbf{\Pi}_{\mathcal{T}(\boldsymbol{x})^{\perp}} \sum_{n \in [N]} \sqrt{\sigma_n}\langle \tilde{\mu}_g, e_n^{\mathcal{F}} \rangle_{\mathcal{F}} e_n^{\mathcal{F}} = \sum_{n \in [N]} \langle \tilde{\mu}_g, e_n^{\mathcal{F}} \rangle_{\mathcal{F}} \sqrt{\sigma_n} \mathbf{\Pi}_{\mathcal{T}(\boldsymbol{x})^{\perp}} e_n^{\mathcal{F}}. \tag{80}$$

Then

$$\|\mathbf{\Pi}_{\mathcal{T}(\boldsymbol{x})^\perp}\boldsymbol{\Sigma}_N^{1/2}\tilde{\mu}_g\|_{\mathcal{F}}^2 = \|\sum_{n\in[N]}\langle\tilde{\mu}_g,e_n^{\mathcal{F}}\rangle_{\mathcal{F}}\sqrt{\sigma_n}\mathbf{\Pi}_{\mathcal{T}(\boldsymbol{x})^\perp}e_n^{\mathcal{F}}\|_{\mathcal{F}}^2 \tag{81}$$

$$= \sum_{n\in[N]}\sum_{m\in[N]}\langle\tilde{\mu}_g,e_n^{\mathcal{F}}\rangle_{\mathcal{F}}\langle\tilde{\mu}_g,e_m^{\mathcal{F}}\rangle_{\mathcal{F}}\sqrt{\sigma_n}\sqrt{\sigma_m}\langle\mathbf{\Pi}_{\mathcal{T}(\boldsymbol{x})^\perp}e_n^{\mathcal{F}},\mathbf{\Pi}_{\mathcal{T}(\boldsymbol{x})^\perp}e_m^{\mathcal{F}}\rangle_{\mathcal{F}}$$

$$\leq \sum_{n\in[N]}\sum_{m\in[N]}\langle\tilde{\mu}_g,e_n^{\mathcal{F}}\rangle_{\mathcal{F}}\langle\tilde{\mu}_g,e_m^{\mathcal{F}}\rangle_{\mathcal{F}}\sqrt{\sigma_n}\sqrt{\sigma_m}\|\mathbf{\Pi}_{\mathcal{T}(\boldsymbol{x})^\perp}e_n^{\mathcal{F}}\|_{\mathcal{F}}\|\mathbf{\Pi}_{\mathcal{T}(\boldsymbol{x})^\perp}e_m^{\mathcal{F}}\|_{\mathcal{F}}$$

$$\leq \left(\sum_{n\in[N]}\sum_{m\in[N]}|\langle\tilde{\mu}_g,e_n^{\mathcal{F}}\rangle_{\mathcal{F}}|\cdot|\langle\tilde{\mu}_g,e_m^{\mathcal{F}}\rangle_{\mathcal{F}}|\right)\max_{n\in[N]}\sigma_n\|\mathbf{\Pi}_{\mathcal{T}(\boldsymbol{x})^\perp}e_n^{\mathcal{F}}\|_{\mathcal{F}}^2$$

$$\leq \left(\sum_{n\in[N]}|\langle\tilde{\mu}_g,e_n^{\mathcal{F}}\rangle_{\mathcal{F}}|\right)^2\max_{n\in[N]}\sigma_n\|\mathbf{\Pi}_{\mathcal{T}(\boldsymbol{x})^\perp}e_n^{\mathcal{F}}\|_{\mathcal{F}}^2. \tag{82}$$

Remarking that $\|g\|_{d\omega,1} = \sum_{n\in[N]}|\langle\tilde{\mu}_g,e_n^{\mathcal{F}}\rangle_{\mathcal{F}}|$ concludes the proof of (17) and therefore Lemma 1. $\square$

### D.3 Proof of Proposition 4

Proposition 4 gives an upper bound to the term $\max_{n\in[N]}\sigma_n\|\mathbf{\Pi}_{\mathcal{T}(\boldsymbol{x})^\perp}e_n^{\mathcal{F}}\|_{\mathcal{F}}^2$ that appears in Lemma 1.
We first prove a technical result, Lemma 4, and then combine it with Proposition 6 to finish the proof.
We conclude with the proof of Proposition 6.

#### D.3.1 A preliminary lemma

Let $\boldsymbol{x} = (x_1,\ldots,x_N) \in \mathcal{X}^N$. Recall that $\boldsymbol{K}(\boldsymbol{x}) = (k(x_i,x_j))_{1\leq i,j\leq N}$ and denote $\tilde{\boldsymbol{K}}(\boldsymbol{x}) = (\tilde{k}(x_i,x_j))_{1\leq i,j\leq N}$, see section 4.2.2. In the following, we define

$$\Delta_n^{\mathcal{F}}(\boldsymbol{x}) = e_n^{\mathcal{F}}(\boldsymbol{x})^\intercal\boldsymbol{K}(\boldsymbol{x})^{-1}e_n^{\mathcal{F}}(\boldsymbol{x}) \tag{83}$$

$$\Delta_n^{\tilde{\mathcal{F}}}(\boldsymbol{x}) = e_n^{\tilde{\mathcal{F}}}(\boldsymbol{x})^\intercal\tilde{\boldsymbol{K}}(\boldsymbol{x})^{-1}e_n^{\tilde{\mathcal{F}}}(\boldsymbol{x}) \tag{84}$$

Lemma 4 below shows that each term of the form $\Delta_n^{\mathcal{F}}(\boldsymbol{x})$ measures the squared norm of the projection of $e_n^{\mathcal{F}}$ on $\mathcal{T}(\boldsymbol{x})$. The same holds for $\Delta_n^{\tilde{\mathcal{F}}}(\boldsymbol{x})$ and the projection of $e_n^{\tilde{\mathcal{F}}}$ onto $\tilde{\mathcal{T}}(\boldsymbol{x})$.

Indeed, $\|\mathbf{\Pi}_{\mathcal{T}(\boldsymbol{x})^\perp}e_n^{\mathcal{F}}\|_{\mathcal{F}}^2 = 1 - \|\mathbf{\Pi}_{\mathcal{T}(\boldsymbol{x})}e_n^{\mathcal{F}}\|_{\mathcal{F}}^2$ since $\|e_n^{\mathcal{F}}\|_{\mathcal{F}}^2 = 1$. Thus it is sufficient to prove that $\|\mathbf{\Pi}_{\mathcal{T}(\boldsymbol{x})}e_n^{\mathcal{F}}\|_{\mathcal{F}}^2 = \Delta_n^{\mathcal{F}}(\boldsymbol{x})$. This boils down to showing that $\boldsymbol{K}(\boldsymbol{x})^{-1}$ is the matrix of the inner product $\langle\cdot,\cdot\rangle_{\mathcal{F}}$ restricted to $\mathcal{T}(\boldsymbol{x})$.

**Lemma 4.** *For $n \in \mathbb{N}^*$, let $e_n^{\mathcal{F}}(\boldsymbol{x}), e_n^{\tilde{\mathcal{F}}}(\boldsymbol{x}) \in \mathbb{R}^N$ the vectors of the evaluations of $e_n^{\mathcal{F}}$ and $e_n^{\tilde{\mathcal{F}}}$ on the elements of $\boldsymbol{x}$ respectively. Then*

$$\|\mathbf{\Pi}_{\mathcal{T}(\boldsymbol{x})^\perp}e_n^{\mathcal{F}}\|_{\mathcal{F}}^2 = 1 - \Delta_n^{\mathcal{F}}(\boldsymbol{x}), \tag{85}$$

$$\|\mathbf{\Pi}_{\tilde{\mathcal{T}}_N(\boldsymbol{x})^\perp}e_n^{\tilde{\mathcal{F}}}\|_{\tilde{\mathcal{F}}}^2 = 1 - \Delta_n^{\tilde{\mathcal{F}}}(\boldsymbol{x}). \tag{86}$$

We give the proof of (85); the proof of (86) follows the same lines.

*Proof.* Let us write

$$\mathbf{\Pi}_{\mathcal{T}(\boldsymbol{x})}e_n^{\mathcal{F}} = \sum_{i\in[N]}c_ik(x_i,.), \tag{87}$$

where the $c_i$ are the elements of the vector $\boldsymbol{c} = \boldsymbol{K}(\boldsymbol{x})^{-1}e_n^{\mathcal{F}}(\boldsymbol{x})$. Then

$$\mathbf{\Pi}_{\mathcal{T}(\boldsymbol{x})}e_n^{\mathcal{F}} = \sum_{i\in[N]}c_i\sum_{m\in\mathbb{N}^*}\sigma_me_m(x_i)e_m(.) \tag{88}$$

$$= \sum_{m\in\mathbb{N}^*}\sqrt{\sigma_m}\left(\sum_{i\in[N]}c_ie_m(x_i)\right)e_m^{\mathcal{F}}(.).$$

Since $(e_m^{\mathcal{F}})_{m \in \mathbb{N}^*}$ is orthonormal,

$$
\begin{aligned}
\|\mathbf{\Pi}_{\mathcal{T}(\boldsymbol{x})} e_n^{\mathcal{F}}\|_{\mathcal{F}}^2 &= \sum_{m \in \mathbb{N}^*} \sigma_m \left( \sum_{i \in [N]} c_i e_m(x_i) \right)^2 \qquad (89) \\
&= \sum_{m \in \mathbb{N}^*} \sigma_m \sum_{i \in [N]} \sum_{j \in [N]} c_i c_j e_m(x_i) e_m(x_j) \\
&= \sum_{m \in \mathbb{N}^*} \boldsymbol{c}^\mathsf{T} e_m^{\mathcal{F}}(\boldsymbol{x}) e_m^{\mathcal{F}}(\boldsymbol{x})^\mathsf{T} \boldsymbol{c} \\
&= \boldsymbol{c}^\mathsf{T} \sum_{m \in \mathbb{N}^*} e_m^{\mathcal{F}}(\boldsymbol{x}) e_m^{\mathcal{F}}(\boldsymbol{x})^\mathsf{T} \boldsymbol{c}.
\end{aligned}
$$

Using Mercer's theorem, see (56),

$$
\boldsymbol{K}(\boldsymbol{x}) = \sum_{m \in \mathbb{N}^*} e_m^{\mathcal{F}}(\boldsymbol{x}) e_m^{\mathcal{F}}(\boldsymbol{x})^\mathsf{T}. \qquad (90)
$$

Combining (89) and (90) along with the definition of the vector $\boldsymbol{c} = \boldsymbol{K}(\boldsymbol{x})^{-1} e_n^{\mathcal{F}}(\boldsymbol{x})$ yields

$$
\begin{aligned}
\|\mathbf{\Pi}_{\mathcal{T}(\boldsymbol{x})} e_n^{\mathcal{F}}\|_{\mathcal{F}}^2 &= \boldsymbol{c}^\mathsf{T} \boldsymbol{K}(\boldsymbol{x}) \boldsymbol{c} \qquad (91) \\
&= e_n^{\mathcal{F}}(\boldsymbol{x})^\mathsf{T} \boldsymbol{K}(\boldsymbol{x})^{-1} \boldsymbol{K}(\boldsymbol{x}) \boldsymbol{K}(\boldsymbol{x})^{-1} e_n^{\mathcal{F}}(\boldsymbol{x}) \\
&= e_n^{\mathcal{F}}(\boldsymbol{x})^\mathsf{T} \boldsymbol{K}(\boldsymbol{x})^{-1} e_n^{\mathcal{F}}(\boldsymbol{x}) \\
&= \Delta_n^{\mathcal{F}}(\boldsymbol{x}).
\end{aligned}
$$

$\square$

### D.3.2 End of the proof of Proposition 4

*Proof.* By Lemma 4, the inequality (22) in Proposition 4 is equivalent to

$$
\forall n \in [N], \ \sigma_n \left( 1 - \Delta_n^{\mathcal{F}}(\boldsymbol{x}) \right) \leq \sigma_1 \left( 1 - \Delta_n^{\tilde{\mathcal{F}}}(\boldsymbol{x}) \right). \qquad (92)
$$

As an intermediate remark, note that in the special case $n = 1$, by construction

$$
\boldsymbol{K}(\boldsymbol{x}) \prec \tilde{\boldsymbol{K}}(\boldsymbol{x}), \qquad (93)
$$

where $\prec$ is the Loewner order, the partial order defined by the convex cone of positive semi-definite matrices. Thus

$$
\tilde{\boldsymbol{K}}(\boldsymbol{x})^{-1} \prec \boldsymbol{K}(\boldsymbol{x})^{-1}. \qquad (94)
$$

Noting that $\tilde{\sigma}_1 = \sigma_1$ and that

$$
e_1^{\mathcal{F}} = \sqrt{\sigma_1} e_1 = \sqrt{\tilde{\sigma}_1} e_1 = e_1^{\tilde{\mathcal{F}}}. \qquad (95)
$$

yields (92) for $n = 1$:

$$
1 - e_1^{\mathcal{F}}(\boldsymbol{x})^\mathsf{T} \boldsymbol{K}(\boldsymbol{x})^{-1} e_1^{\mathcal{F}}(\boldsymbol{x}) \leq 1 - e_1^{\tilde{\mathcal{F}}}(\boldsymbol{x})^\mathsf{T} \tilde{\boldsymbol{K}}(\boldsymbol{x})^{-1} e_1^{\tilde{\mathcal{F}}}(\boldsymbol{x}). \qquad (96)
$$

For $n \neq 1$, the proof is much more subtle. Indeed, a naive application of the inequality (94) would lead to the following inequality

$$
1 - e_n^{\tilde{\mathcal{F}}}(\boldsymbol{x})^\mathsf{T} \boldsymbol{K}(\boldsymbol{x})^{-1} e_n^{\tilde{\mathcal{F}}}(\boldsymbol{x}) \leq 1 - e_n^{\tilde{\mathcal{F}}}(\boldsymbol{x})^\mathsf{T} \tilde{\boldsymbol{K}}(\boldsymbol{x})^{-1} e_n^{\tilde{\mathcal{F}}}(\boldsymbol{x}). \qquad (97)
$$

Since $\forall n \in \mathbb{N}, e_n^{\tilde{\mathcal{F}}} = \sqrt{\sigma_1/\sigma_n} e_n^{\mathcal{F}}$, we get

$$
1 - \sigma_1 e_n^{\mathcal{F}}(\boldsymbol{x})^\mathsf{T} \boldsymbol{K}(\boldsymbol{x})^{-1} e_n^{\mathcal{F}}(\boldsymbol{x}) \leq 1 - \sigma_n e_n^{\tilde{\mathcal{F}}}(\boldsymbol{x})^\mathsf{T} \tilde{\boldsymbol{K}}(\boldsymbol{x})^{-1} e_n^{\tilde{\mathcal{F}}}(\boldsymbol{x}), \qquad (98)
$$

and hence the unsatisfactory inequality

$$
1 - \sigma_1 \Delta_n^{\mathcal{F}}(\boldsymbol{x}) \leq 1 - \sigma_n \Delta_n^{\tilde{\mathcal{F}}}(\boldsymbol{x}) \qquad (99)
$$

We can prove a better inequality by applying a sequence of rank-one updates to the kernel $k$ to build $N$ intermediate kernels $k^{(\ell)}$ that lead to $N$ inequalities sharp enough to prove (92) for $n \neq 1$. Then inequality (92) will result as a corollary of Proposition 6 below. To this aim, we define $N$ RKHS $\tilde{\mathcal{F}}_\ell$, $1 \leq \ell \leq N$, that interpolate between $\mathcal{F}$ and $\tilde{\mathcal{F}}$. For $\ell \in [N]$, define the kernel $\tilde{k}^{(\ell)}$ by

$$\tilde{k}^{(\ell)}(x,y) = \sum_{m \in [\ell]} \sigma_1 e_m(x) e_m(y) + \sum_{m \geq \ell+1} \sigma_m e_m(x) e_m(y), \tag{100}$$

and let $\tilde{\mathcal{F}}_\ell$ the RKHS corresponding to the kernel $\tilde{k}^{(\ell)}$. For $\boldsymbol{x} \in \mathcal{X}^N$, define $\tilde{\boldsymbol{K}}^{(\ell)}(\boldsymbol{x}) = (\tilde{k}^{(\ell)}(x_i, x_j))_{1 \leq i,j \leq N}$. Similar to previous notations, we define as well

$$\Delta_n^{\tilde{\mathcal{F}}_\ell}(\boldsymbol{x}) = e_n^{\tilde{\mathcal{F}}_\ell}(\boldsymbol{x})^\intercal \tilde{\boldsymbol{K}}^{(\ell)}(\boldsymbol{x})^{-1} e_n^{\tilde{\mathcal{F}}_\ell}(\boldsymbol{x}). \tag{101}$$

Now we have the following useful proposition.

**Proposition 6.** *For $n \in [N] \setminus \{1\}$, we have*

$$\sigma_n \left( 1 - \Delta_n^{\tilde{\mathcal{F}}_{n-1}}(\boldsymbol{x}) \right) \leq \sigma_1 \left( 1 - \Delta_n^{\tilde{\mathcal{F}}_n}(\boldsymbol{x}) \right), \tag{102}$$

*and*

$$\forall \ell \in [N] \setminus \{1, n\}, \ 1 - \Delta_n^{\tilde{\mathcal{F}}_{\ell-1}}(\boldsymbol{x}) \leq 1 - \Delta_n^{\tilde{\mathcal{F}}_\ell}(\boldsymbol{x}). \tag{103}$$

For ease of reading, we first show that inequality (92) and therefore Proposition 4 is easily deduced from this Proposition 6 and then give its proof.

Let $n \in [N]$ such that $n \neq 1$. We first remark that $\mathcal{F} = \tilde{\mathcal{F}}_1$ and use $(n-2)$ times inequality (103) of Proposition 6:

$$\sigma_n \left( 1 - \Delta_n^{\mathcal{F}}(\boldsymbol{x}) \right) = \sigma_n \left( 1 - \Delta_n^{\tilde{\mathcal{F}}_1}(\boldsymbol{x}) \right) \tag{104}$$
$$\leq \sigma_n \left( 1 - \Delta_n^{\tilde{\mathcal{F}}_{n-1}}(\boldsymbol{x}) \right)$$

Then we use (102) that is connected to the rank-one update from the kernel $k^{(n-1)}$ to $k^{(n)}$ so that

$$\sigma_n \left( 1 - \Delta_n^{\tilde{\mathcal{F}}_{n-1}}(\boldsymbol{x}) \right) \leq \sigma_1 \left( 1 - \Delta_n^{\tilde{\mathcal{F}}_n}(\boldsymbol{x}) \right) \tag{105}$$

Then we apply (103) to the r.h.s. again $N - n - 1$ times to finally get:

$$\sigma_n \left( 1 - \Delta_n^{\mathcal{F}}(\boldsymbol{x}) \right) \leq \sigma_1 \left( 1 - \Delta_n^{\tilde{\mathcal{F}}_N}(\boldsymbol{x}) \right) \tag{106}$$
$$\leq \sigma_1 \left( 1 - \Delta_n^{\tilde{\mathcal{F}}}(\boldsymbol{x}) \right),$$

since $\tilde{k}^{(N)} = \tilde{k}$ and $\tilde{\mathcal{F}}_N = \tilde{\mathcal{F}}$. This concludes the proof of the desired inequality (92) and therefore of Proposition 4. □

### D.3.3 Proof of Proposition 6

*Proof.* (Proposition 6) Let $n \in [N] \setminus \{1\}$, and $M \in \mathbb{N}$ such that $M \geq N$. Let $\boldsymbol{A}_\ell \in \mathbb{R}^{N \times M}$ defined by

$$\forall (i,m) \in [N] \times [M], \ (\boldsymbol{A}_\ell)_{i,m} = e_m^{\tilde{\mathcal{F}}_\ell}(x_i).^{[1]} \tag{107}$$

For $\ell \in [N]$ define

$$\tilde{\boldsymbol{K}}_M^{(\ell)}(\boldsymbol{x}) = \boldsymbol{A}_\ell^\intercal \boldsymbol{A}_\ell. \tag{108}$$

Let $\boldsymbol{W}_\ell \in \mathbb{R}^{M \times M}$ the diagonal matrix defined by

$$\boldsymbol{W}_\ell = \mathrm{diag}(\underbrace{1, ..., 1}_{\ell-1}, \sqrt{\frac{\sigma_1}{\sigma_\ell}}, 1..., 1) \tag{109}$$

Then one has the simple relation

$$\boldsymbol{A}_{\ell+1} = \boldsymbol{A}_\ell \boldsymbol{W}_\ell, \tag{110}$$

which prepares the use of Lemma 3 in Section C.2. By definition of the $n$-th leverage score of the matrix $\boldsymbol{A}$, see (31) in Section C.2,

$$e_n^{\tilde{\mathcal{F}}_\ell}(\boldsymbol{x})^\intercal \tilde{\boldsymbol{K}}_M^{(\ell)}(\boldsymbol{x})^{-1} e_n^{\tilde{\mathcal{F}}_\ell}(\boldsymbol{x}) = e_n^{\tilde{\mathcal{F}}_\ell}(\boldsymbol{x})^\intercal (\boldsymbol{A}_\ell^\intercal \boldsymbol{A}_\ell)^{-1} e_n^{\tilde{\mathcal{F}}_\ell}(\boldsymbol{x}) = \tau_n (\boldsymbol{A}_\ell). \tag{111}$$

Define similarly $\Delta_{n,M}^{\tilde{\mathcal{F}}_\ell}(\boldsymbol{x}) = e_n^{\tilde{\mathcal{F}}_\ell}(\boldsymbol{x})^\intercal \tilde{\boldsymbol{K}}_M^{(\ell)}(\boldsymbol{x})^{-1} e_n^{\tilde{\mathcal{F}}_\ell}(\boldsymbol{x})$. Thanks to (34) of Lemma 3 and (110) and for $\ell = n$

$$\tau_n \Big( \boldsymbol{A}_n \Big) = \tau_n (\boldsymbol{A}_{n-1} \boldsymbol{W}_n) = \frac{(1+\rho_n)\tau_n \Big( \boldsymbol{A}_{n-1} \Big)}{1 + \rho_n \tau_n \Big( \boldsymbol{A}_{n-1} \Big)}, \tag{112}$$

where $\rho_n = \dfrac{\sigma_1}{\sigma_n} - 1$. Thus

$$1 - \tau_n \Big( \boldsymbol{A}_n \Big) = 1 - \frac{(1+\rho_n)\tau_n \Big( \boldsymbol{A}_{n-1} \Big)}{1 + \rho_n \tau_n \Big( \boldsymbol{A}_{n-1} \Big)} = \frac{1 - \tau_n \Big( \boldsymbol{A}_{n-1} \Big)}{1 + \rho_n \tau_n \Big( \boldsymbol{A}_{n-1} \Big)}. \tag{113}$$

Then

$$\sigma_1 \left( 1 - \tau_n \Big( \boldsymbol{A}_n \Big) \right) = \sigma_1 \frac{1 - \tau_n \Big( \boldsymbol{A}_{n-1} \Big)}{1 + \rho_n \tau_n \Big( \boldsymbol{A}_{n-1} \Big)} \tag{114}$$

$$= \sigma_n (1+\rho_n) \frac{1 - \tau_n \Big( \boldsymbol{A}_{n-1} \Big)}{1 + \rho_n \tau_n \Big( \boldsymbol{A}_{n-1} \Big)}$$

$$= \frac{1+\rho_n}{1 + \rho_n \tau_n \Big( \boldsymbol{A}_{n-1} \Big)} \sigma_n \left( 1 - \tau_n \Big( \boldsymbol{A}_{n-1} \Big) \right)$$

$$\geq \sigma_n \left( 1 - \tau_n \Big( \boldsymbol{A}_{n-1} \Big) \right),$$

since $\rho_n \geq 0$ and $\tau_n \Big( \boldsymbol{A}_{n-1} \Big) \in [0,1]$ thanks to (33). This proves that for $M \in \mathbb{N}^*$ such that $M \geq N$,

$$\sigma_n \left( 1 - \Delta_{n,M}^{\tilde{\mathcal{F}}_{n-1}}(\boldsymbol{x}) \right) \leq \sigma_1 \left( 1 - \Delta_{n,M}^{\tilde{\mathcal{F}}_n}(\boldsymbol{x}) \right). \tag{115}$$

Now,

$$\lim_{M \to \infty} \tilde{\boldsymbol{K}}_M^{(n+1)}(\boldsymbol{x}) = \tilde{\boldsymbol{K}}^{(n+1)}(\boldsymbol{x}), \tag{116}$$

$$\lim_{M \to \infty} \tilde{\boldsymbol{K}}_M^{(n)}(\boldsymbol{x}) = \tilde{\boldsymbol{K}}^{(n)}(\boldsymbol{x}). \tag{117}$$

Moreover the application $\boldsymbol{X} \mapsto \boldsymbol{X}^{-1}$ is continuous in $GL_N(\mathbb{R})$. This proves the inequality (102) of Proposition 6. To prove the inequality (103), we start by using (35):

$$\forall \ell \in [N] \setminus \{1, n\}, \ \tau_n \Big( \boldsymbol{A}(\ell) \Big) = \tau_n (\boldsymbol{A}_{\ell-1} \boldsymbol{W}_\ell) \leq \tau_n \Big( \boldsymbol{A}_{\ell-1} \Big). \tag{118}$$

which implies that

$$\forall \ell \in [N] \setminus \{1, n\}, 1 - \tau_n \Big( \boldsymbol{A}_{\ell-1} \Big) \leq 1 - \tau_n \Big( \boldsymbol{A}_\ell \Big). \tag{119}$$

Then for $M \geq N$,

$$\forall \ell \in [N] \setminus \{1, n\}, \ 1 - \Delta_{n,M}^{\tilde{\mathcal{F}}_{\ell-1}}(\boldsymbol{x}) \leq 1 - \Delta_{n,M}^{\tilde{\mathcal{F}}_\ell}(\boldsymbol{x}). \tag{120}$$

As above, we conclude the proof by considering the limit $M \to \infty$

$$\forall \ell \in [N] \setminus \{1, n\}, \ \lim_{M \to \infty} \tilde{\boldsymbol{K}}_M^{(\ell)}(\boldsymbol{x}) = \tilde{\boldsymbol{K}}^{(\ell)}(\boldsymbol{x}). \tag{121}$$

This proves inequality (103) and concludes the proof of Proposition 6. $\square$

## D.4 Proof of Proposition 3

In this section, $\boldsymbol{x} = (x_1, \ldots, x_N) \in \mathcal{X}^N$ is the realization of the DPP of Theorem 1. Let $\boldsymbol{E}^{\mathcal{F}}(\boldsymbol{x}) = (e_i^{\mathcal{F}}(x_j))_{1 \leq i,j \leq N}$ and $\boldsymbol{E}(\boldsymbol{x}) = (e_i(x_j))_{1 \leq i,j \leq N}$, and $\boldsymbol{K}(\boldsymbol{x}) = (k(x_i, x_j))_{1 \leq i,j \leq N}$. Moreover, let $\mathcal{E}_N^{\mathcal{F}} = \mathrm{Span}(e_m^{\mathcal{F}})_{m \in [N]}$ and $\mathcal{T}(\boldsymbol{x}) = \mathrm{Span}\,(k(x_i, .))_{i \in [N]}$.

We first prove two lemmas that are necessary to prove Proposition 3.

### D.4.1 Two preliminary lemmas

**Lemma 5.** *Let $\boldsymbol{x} = (x_1, \ldots, x_N) \in \mathcal{X}^N$ such that $\mathrm{Det}^2 \boldsymbol{E}(\boldsymbol{x}) \neq 0$. Then,*

$$\prod_{\ell \in [N]} \frac{1}{\cos^2 \theta_\ell \left(\mathcal{E}_N^{\mathcal{F}}, \mathcal{T}(\boldsymbol{x})\right)} = \frac{\mathrm{Det}\, \boldsymbol{K}(\boldsymbol{x})}{\mathrm{Det}^2 \boldsymbol{E}^{\mathcal{F}}(\boldsymbol{x})}. \tag{122}$$

*Proof.* The condition $\mathrm{Det}^2 \boldsymbol{E}(\boldsymbol{x}) \neq 0$ yields by Proposition 2 that $\boldsymbol{K}(\boldsymbol{x})$ is non singular. Thus $\dim \mathcal{T}(\boldsymbol{x}) = N$. Let $(t_i)_{i \in [N]}$ an orthonormal basis of $\mathcal{T}(\boldsymbol{x})$ with respect to $\langle ., . \rangle_{\mathcal{F}}$. Using Corollary 2, and the fact that $(e_n^{\mathcal{F}})_{n \in [N]}$ is an orthonormal basis of $\mathcal{E}_N^{\mathcal{F}}$ according to $\langle ., . \rangle_{\mathcal{F}}$,

$$\prod_{\ell \in [N]} \cos^2 \theta_\ell \left(\mathcal{E}_N^{\mathcal{F}}, \mathcal{T}(\boldsymbol{x})\right) = \mathrm{Det}^2(\langle e_n^{\mathcal{F}}, t_i \rangle_{\mathcal{F}})_{(n,i) \in [N] \times [N]}. \tag{123}$$

Now, write for $i \in [N]$,

$$t_i = \sum_{j \in [N]} c_{i,j} k(x_j, .). \tag{124}$$

Thus

$$\langle e_n^{\mathcal{F}}, t_i \rangle_{\mathcal{F}} = \sum_{j \in [N]} c_{i,j} \langle e_n^{\mathcal{F}}, k(x_j, .) \rangle_{\mathcal{F}} \tag{125}$$

$$= \sum_{j \in [N]} c_{i,j} e_n^{\mathcal{F}}(x_j). \tag{126}$$

Then

$$(\langle e_n^{\mathcal{F}}, t_i \rangle_{\mathcal{F}})_{(n,i) \in [N] \times [N]} = \boldsymbol{E}^{\mathcal{F}}(\boldsymbol{x}) \boldsymbol{C}(\boldsymbol{x})^{\mathsf{T}}, \tag{127}$$

where

$$\boldsymbol{C}(\boldsymbol{x}) = (c_{i,j})_{1 \leq i,j \leq N}. \tag{128}$$

Thus

$$\mathrm{Det}^2(\langle e_n^{\mathcal{F}}, t_i \rangle_{\mathcal{F}})_{(n,i) \in [N] \times [N]} = \mathrm{Det}^2 \boldsymbol{C}(\boldsymbol{x}) \, \mathrm{Det}^2 \boldsymbol{E}^{\mathcal{F}}(\boldsymbol{x}). \tag{129}$$

Now, let $\boldsymbol{c}_i$ the columns of the matrix $\boldsymbol{C}(\boldsymbol{x})$. $(t_i)_{i \in [N]}$ is an orthonormal basis of $\mathcal{T}(\boldsymbol{x})$ with respect to $\langle ., . \rangle_{\mathcal{F}}$, then by (124)

$$\delta_{i,i'} = \langle t_i, t_{i'} \rangle_{\mathcal{F}} = \boldsymbol{c}_i^{\mathsf{T}} \boldsymbol{K}(\boldsymbol{x}) \boldsymbol{c}_{i'}. \tag{130}$$

Therefore

$$\boldsymbol{C}(\boldsymbol{x})^{\mathsf{T}} \boldsymbol{K}(\boldsymbol{x}) \boldsymbol{C}(\boldsymbol{x}) = \mathbb{I}_N. \tag{131}$$

Thus

$$\mathrm{Det}^2 \boldsymbol{C}(\boldsymbol{x}) = \frac{1}{\mathrm{Det}\, \boldsymbol{K}(\boldsymbol{x})}. \tag{132}$$

Combining (123), (129) and (132) concludes the proof of Lemma 5:

$$\prod_{\ell \in [N]} \frac{1}{\cos^2 \theta_\ell \left(\mathcal{E}_N^{\mathcal{F}}, \mathcal{T}(\boldsymbol{x})\right)} = \frac{\mathrm{Det}\, \boldsymbol{K}(\boldsymbol{x})}{\mathrm{Det}^2 \boldsymbol{E}^{\mathcal{F}}(\boldsymbol{x})}. \tag{133}$$

$\square$

**Lemma 6.**

$$\frac{1}{N!} \int_{\mathcal{X}^N} \mathrm{Det}\, \boldsymbol{K}(x_1, \ldots, x_N) \otimes_{j \in [N]} \mathrm{d}\omega(x_j) = \sum_{\substack{T \subset \mathbb{N}^* \\ |T| = N}} \prod_{t \in T} \sigma_t. \tag{134}$$

*Proof.* Let $\boldsymbol{x} = (x_1, \ldots, x_N) \in \mathcal{X}^N$. From (56)

$$\operatorname{Det} \boldsymbol{K}(\boldsymbol{x}) = \lim_{M \to \infty} \operatorname{Det} \boldsymbol{K}_M(\boldsymbol{x}). \tag{135}$$

Moreover,

$$\operatorname{Det} \boldsymbol{K}_M(\boldsymbol{x}) = \sum_{T \subset [M], |T| = N} \prod_{i \in T} \sigma_i \operatorname{Det}^2(e_i(x_j))_{(i,j) \in T \times [N]}. \tag{136}$$

Now, for $T \subset [M]$ such that $|T| = N$, $(e_t)_{t \in T}$ is an orthonormal family of $\mathbb{L}_2(d\omega)$, then by [11] Lemma 21:

$$\int_{\mathcal{X}^N} \operatorname{Det}^2(e_t(x_j)) \otimes_{j \in [N]} d\omega(x_j) = N!. \tag{137}$$

Thus

$$\frac{1}{N!} \int_{\mathcal{X}^N} \operatorname{Det} \boldsymbol{K}_M(\boldsymbol{x}) \otimes_{j \in [N]} d\omega(x_j) = \frac{1}{N!} \sum_{T \subset [M], |T| = N} \prod_{t \in T} \sigma_t \int_{\mathcal{X}^N} \operatorname{Det}^2(e_t(x_j)) \otimes_{j \in [N]} d\omega(x_j) \tag{138}$$

$$= \sum_{T \subset [M], |T| = N} \prod_{t \in T} \sigma_t.$$

Now, $\sum_{n \in \mathbb{N}^*} \sigma_n < \infty$ implies that $\sum_{T \subset \mathbb{N}^*, |T| = N} \prod_{t \in T} \sigma_t < \infty$. In fact, for $\ell \in [N]$ let $p_\ell$ the $\ell$-th symmetric polynomial. By Maclaurin's inequality [22], and for any vector $\boldsymbol{\nu} \in \mathbb{R}_+^M$

$$\left( \frac{p_\ell(\boldsymbol{\nu})}{\binom{M}{\ell}} \right)^{\frac{1}{\ell}} \leq \frac{p_1(\boldsymbol{\nu})}{M}. \tag{139}$$

Thus

$$p_\ell(\boldsymbol{\nu}) \leq \frac{\binom{M}{\ell}}{M^\ell} p_1(\boldsymbol{\nu})^\ell \tag{140}$$

$$\leq \frac{M!}{\ell!(M - \ell)!M^\ell} p_1(\boldsymbol{\nu})^\ell$$

$$\leq \frac{M(M - 1) \ldots (M - \ell + 1)}{\ell! M^\ell} p_1(\boldsymbol{\nu})^\ell$$

$$\leq \frac{1}{\ell!} p_1(\boldsymbol{\nu})^\ell.$$

This inequality is independent of the dimension $M$ thus it can be extended for $\boldsymbol{\nu} \in \mathbb{R}_+^{\mathbb{N}^*}$ with $\sum_{n \in \mathbb{N}^*} \nu_n < \infty$. Therefore

$$\sum_{T \subset \mathbb{N}^*, |T| = N} \prod_{t \in T} \sigma_t \leq \frac{1}{N!} \left( \sum_{n \in \mathbb{N}^*} \sigma_n \right)^N < \infty. \tag{141}$$

Furthermore,

$$\forall M \in \mathbb{N}^*, \forall \boldsymbol{x} \in \mathcal{X}^N, 0 \leq \operatorname{Det} \boldsymbol{K}_M(\boldsymbol{x}) \leq \operatorname{Det} \boldsymbol{K}_{M+1,N}(\boldsymbol{x}). \tag{142}$$

Then by monotone convergence theorem, $\boldsymbol{x} \mapsto \frac{1}{N!} \operatorname{Det} \boldsymbol{K}(\boldsymbol{x})$ is mesurable and

$$\int_{\mathcal{X}^N} \frac{1}{N!} \operatorname{Det} \boldsymbol{K}(\boldsymbol{x}) \otimes_{j \in [N]} d\omega(x_j) = \lim_{M \to \infty} \int_{\mathcal{X}^N} \frac{1}{N!} \operatorname{Det} \boldsymbol{K}_M(\boldsymbol{x}) \otimes_{j \in [N]} d\omega(x_j) \tag{143}$$

$$= \lim_{M \to \infty} \sum_{T \subset [M], |T| = N} \prod_{t \in T} \sigma_t$$

$$= \sum_{T \subset \mathbb{N}^*, |T| = N} \prod_{t \in T} \sigma_t.$$

$\square$

### D.4.2  End of the proof of Proposition 3

*Proof.* Remember that

$$\mathbb{P}\left(\mathrm{Det}\,\boldsymbol{E}(\boldsymbol{x})\neq 0\right)=1. \tag{144}$$

Then by Lemma 5 and the fact that $\mathrm{Det}^2\,\boldsymbol{E}^{\mathcal{F}}(\boldsymbol{x})=\prod\limits_{n\in[N]}\sigma_n\,\mathrm{Det}^2\,\boldsymbol{E}(\boldsymbol{x})$

$$\prod_{\ell\in[N]}\frac{1}{\cos^2\theta_\ell\left(\mathcal{E}_N^{\mathcal{F}},\mathcal{T}(\boldsymbol{x})\right)}=\frac{\mathrm{Det}\,\boldsymbol{K}(\boldsymbol{x})}{\mathrm{Det}^2\,\boldsymbol{E}^{\mathcal{F}}(\boldsymbol{x})}=\frac{1}{\prod\limits_{n\in[N]}\sigma_n}\frac{\mathrm{Det}\,\boldsymbol{K}(\boldsymbol{x})}{\mathrm{Det}^2\,\boldsymbol{E}(\boldsymbol{x})}. \tag{145}$$

Then, taking the expectation with respect to $\boldsymbol{x}$ resulting from a DPP of kernel $\mathfrak{K}(x,y)$,

$$\mathbb{E}_{\mathrm{DPP}}\prod_{\ell\in[N]}\frac{1}{\cos^2\theta_\ell\left(\mathcal{E}_N^{\mathcal{F}},\mathcal{T}(\boldsymbol{x})\right)}=\frac{1}{N!}\int_{\mathcal{X}^N}\mathrm{Det}^2\,\boldsymbol{E}(\boldsymbol{x})\prod_{\ell\in[N]}\frac{1}{\cos^2\theta_\ell\left(\mathcal{E}_N^{\mathcal{F}},\mathcal{T}(\boldsymbol{x})\right)}\otimes_{i=1}^N\,\mathrm{d}\omega(x_i) \tag{146}$$

$$=\frac{1}{N!}\int_{\mathcal{X}^N}\mathrm{Det}^2\,\boldsymbol{E}(\boldsymbol{x})\frac{1}{\prod\limits_{n\in[N]}\sigma_n}\frac{\mathrm{Det}\,\boldsymbol{K}(\boldsymbol{x})}{\mathrm{Det}^2\,\boldsymbol{E}(\boldsymbol{x})}\otimes_{i=1}^N\,\mathrm{d}\omega(x_i)$$

$$=\frac{1}{\prod\limits_{n\in[N]}\sigma_n}\frac{1}{N!}\int_{\mathcal{X}^N}\mathrm{Det}\,\boldsymbol{K}(\boldsymbol{x})\otimes_{i=1}^N\,\mathrm{d}\omega(x_i).$$

Now, by Lemma 6

$$\frac{1}{N!}\int_{\mathcal{X}^N}\mathrm{Det}\,\boldsymbol{K}(\boldsymbol{x})\otimes_{i=1}^N\,\mathrm{d}\omega(x_i)=\sum_{\substack{T\subset\mathbb{N}^*\\|T|=N}}\prod_{t\in T}\sigma_t. \tag{147}$$

Therefore,

$$\mathbb{E}_{\mathrm{DPP}}\prod_{\ell\in[N]}\frac{1}{\cos^2\theta_\ell\left(\mathcal{E}_N^{\mathcal{F}},\mathcal{T}(\boldsymbol{x})\right)}=\sum_{\substack{T\subset\mathbb{N}^*\\|T|=N}}\frac{\prod\limits_{t\in T}\sigma_t}{\prod\limits_{n\in[N]}\sigma_n}. \tag{148}$$

$\square$

### D.5  Proof of Theorem 1

*Proof.* Thanks to Proposition 4 and Lemma 2 (for $\tilde{\mathcal{F}}$ and $\tilde{k}$)

$$\max_{n\in[N]}\sigma_n\|\boldsymbol{\Pi}_{\mathcal{T}(\boldsymbol{x})^\perp}e_n^{\mathcal{F}}\|_{\mathcal{F}}^2\leq\sigma_1\cdot\max_{n\in[N]}\|\boldsymbol{\Pi}_{\tilde{\mathcal{T}}(\boldsymbol{x})^\perp}e_n^{\tilde{\mathcal{F}}}\|_{\tilde{\mathcal{F}}}^2 \tag{149}$$

$$\leq\sigma_1\cdot\left(\prod_{n\in[N]}\frac{1}{\cos^2\theta_n(\tilde{\mathcal{T}}(\boldsymbol{x}),\mathcal{E}_N^{\tilde{\mathcal{F}}})}-1\right). \tag{150}$$

Then Proposition 3 applied to $\tilde{\mathcal{F}}$ with kernel $\tilde{k}$ yields

$$\mathbb{E}_{\mathrm{DPP}}\prod_{n\in[N]}\frac{1}{\cos^2\theta_n\left(\mathcal{E}_N^{\tilde{\mathcal{F}}},\tilde{\mathcal{T}}_N(\boldsymbol{x})\right)}=\sum_{\substack{T\subset\mathbb{N}^*\\|T|=N}}\frac{\prod\limits_{t\in T}\tilde{\sigma}_t}{\prod\limits_{n\in[N]}\tilde{\sigma}_n}. \tag{151}$$

Every subset $T\subset\mathbb{N}^*$ such that $|T|=N$ can be written as $T=V\cup W$ with $V\subset[N]$ and $W\subset\mathbb{N}^*\smallsetminus[N]$, and this decomposition is unique. Then

$$\frac{\prod\limits_{t\in T}\tilde{\sigma}_t}{\prod\limits_{n\in[N]}\tilde{\sigma}_n}=\frac{\prod\limits_{v\in V}\tilde{\sigma}_v\prod\limits_{w\in W}\tilde{\sigma}_w}{\prod\limits_{n\in[N]}\tilde{\sigma}_n}=\frac{\prod\limits_{w\in W}\tilde{\sigma}_w}{\prod\limits_{n\in[N]\smallsetminus V}\tilde{\sigma}_n}. \tag{152}$$

Therefore

$$\sum_{\substack{T\subset\mathbb{N}^* \\ |T|=N}} \frac{\prod\limits_{t\in T}\tilde{\sigma}_t}{\prod\limits_{n\in[N]}\tilde{\sigma}_n} = \sum_{\substack{T\subset\mathbb{N}^* \\ |T|=N \\ T=V\cup W}} \frac{\prod\limits_{w\in W}\tilde{\sigma}_w}{\prod\limits_{n\in[N]\setminus V}\tilde{\sigma}_n} \tag{153}$$

$$= \sum_{V\subset[N]} \sum_{\substack{W\subset\mathbb{N}^*\setminus[N] \\ |W|=N-|V|}} \frac{\prod\limits_{w\in W}\tilde{\sigma}_w}{\prod\limits_{n\in[N]\setminus V}\tilde{\sigma}_n}$$

$$= \sum_{0\leq\ell\leq N}\left[\sum_{\substack{V\subset[N] \\ |V|=\ell}}\prod_{n\in[N]\setminus V}\frac{1}{\tilde{\sigma}_n}\right]\left[\sum_{\substack{W\subset\mathbb{N}^*\setminus[N] \\ |W|=N-\ell}}\prod_{w\in W}\tilde{\sigma}_w\right]$$

$$= \sum_{0\leq\ell\leq N}\left[\sum_{\substack{V\subset[N] \\ |V|=N-\ell}}\prod_{n\in V}\frac{1}{\tilde{\sigma}_n}\right]\left[\sum_{\substack{W\subset\mathbb{N}^*\setminus[N] \\ |W|=N-\ell}}\prod_{w\in W}\tilde{\sigma}_w\right]$$

$$= \sum_{0\leq\ell\leq N} p_{N-\ell}\left(\left(\frac{1}{\tilde{\sigma}_m}\right)_{m\in[N]}\right) p_{N-\ell}\left((\tilde{\sigma}_m)_{m\geq N+1}\right)$$

$$= \sum_{0\leq\ell\leq N} p_{\ell}\left(\left(\frac{1}{\tilde{\sigma}_m}\right)_{m\in[N]}\right) p_{\ell}\left((\tilde{\sigma}_m)_{m\geq N+1}\right),$$

where for $\ell\in[N]$, $p_\ell$ is the $\ell$-th symmetric polynomial with the convention that $p_0=1$.

Finally, thanks to (140) above

$$\sum_{\substack{T\subset\mathbb{N}^* \\ |T|=N}} \frac{\prod\limits_{t\in T}\tilde{\sigma}_t}{\prod\limits_{n\in[N]}\tilde{\sigma}_n} \leq 1 + \sum_{\ell\in[N]}\frac{1}{\ell!^2}\left(\sum_{m\in[N]}\frac{1}{\tilde{\sigma}_m}\sum_{m\geq N+1}\tilde{\sigma}_m\right)^\ell \tag{154}$$

$$\leq 1 + \sum_{\ell\in[N]}\frac{1}{\ell!^2}\left(\frac{N}{\sigma_1}\sum_{m\geq N+1}\sigma_m\right)^\ell.$$

As a consequence, by writing $r_N = \sum\limits_{m\geq N+1}\sigma_m$,

$$\mathbb{E}_{\mathrm{DPP}}\left[\max_{n\in[N]}\sigma_n\|\mathbf{\Pi}_{\mathcal{T}(\boldsymbol{x})^\perp}e_n^{\mathcal{F}}\|_{\mathcal{F}}^2\right] \leq \sigma_1\cdot\sum_{\ell=1}^N\frac{1}{\ell!^2}\left(\frac{Nr_N}{\sigma_1}\right)^\ell \tag{155}$$

which can be plugged in Lemma 1 to conclude the proof. $\qquad\square$

# E  The intuitions behind the algorithm

The algorithm presented in this article is based on several intuitions. In this section, we summarize these intuitions.

## E.1  The geometric intuition

Recall that the quadrature problem in a RKHS boils down to a problem of interpolation of the mean element $\mu_g$ by a mixture of $k(x_i,.)$, where $g\in\mathbb{L}_2(\mathrm{d}\omega)$ such that $\|g\|_{\mathrm{d}\omega}\leq 1$. A promising algorithm would thus be to select the nodes $\{x_i, i\in[N]\}$ so as to minimize the projection of $\mu_g$ onto $\mathcal{T}(\boldsymbol{x}) = \mathrm{Span}(k(x_i,\cdot); i\in[N])$. Upper bounding the approximation error $\|\mu_g - \mathbf{\Pi}_{\mathcal{T}(\boldsymbol{x})}\mu_g\|_{\mathcal{F}}$ is not easy in general. One the one side, we propose to replace $\mu_g$ by its projection $\mathbf{\Pi}_{\mathcal{E}_N^{\mathcal{F}}}\mu_g$ onto the first eigenfunctions of $\boldsymbol{\Sigma}$. Then it is easy to prove that

$$\|\mu_g - \mathbf{\Pi}_{\mathcal{E}_N^{\mathcal{F}}}\mu_g\|_{\mathcal{F}} \leq \sqrt{\sigma_{N+1}}. \tag{156}$$

$$\tilde{\mathcal{T}}(\boldsymbol{x}) = \operatorname{Span} \tilde{k}(x_i, .)_{i \in [N]}$$

$$\mathcal{E}_N^{\tilde{\mathcal{F}}} = \operatorname{Span}(e_j^{\tilde{\mathcal{F}}})_{j \in [N]}$$

$$\theta_N(\tilde{\mathcal{T}}(\boldsymbol{x}), \mathcal{E}_N^{\tilde{\mathcal{F}}})$$

Figure 4: Illustration of the largest principal angle between the subspaces $\tilde{\mathcal{T}}(\boldsymbol{x})$ and $\mathcal{E}_N^{\tilde{\mathcal{F}}}$ in the case of the RKHS of Section 5.1 (the periodic Sobolev space of order 1).

On the other side, if we find a quadrature rule such that $\|\mathbf{\Pi}_{\mathcal{E}_N^{\mathcal{F}}}\mu_g - \mathbf{\Pi}_{\mathcal{T}(\boldsymbol{x})}\mu_g\|_{\mathcal{F}}$ is small, then we can guarantee an overall approximation error that is not too much worse than the PCA error (156). After introducing an auxiliary RKHS $\tilde{\mathcal{F}}$ with kernel $\tilde{k}$, we express this second term using the principal angles between the subspaces $\tilde{\mathcal{T}}(\boldsymbol{x})$ and $\mathcal{E}_N^{\tilde{\mathcal{F}}}$ (see section 4.2.2). This yields a bound on the interpolation error

$$\|\mu_g - \mathbf{\Pi}_{\mathcal{T}(\boldsymbol{x})}\mu_g\|_{\mathcal{F}}^2 \leq 2\left(\sigma_{N+1} + \sigma_1\|g\|_{\mathrm{d}\omega,1}^2 \tan^2\theta_N\left(\mathcal{E}_N^{\tilde{\mathcal{F}}}, \tilde{\mathcal{T}}(\boldsymbol{x})\right)\right). \tag{157}$$

The first term in the right hand side of (157) is $2\sigma_{N+1}$, which corresponds to the approximation error observed in numerical simulations. The second term depends on the largest principal angle $\theta_N$ between the subspaces $\tilde{\mathcal{T}}(\boldsymbol{x})$ and $\mathcal{E}_N^{\tilde{\mathcal{F}}}$, see Figure 4. This term can in turn be bounded by the symmetrized quantity

$$\prod_{\ell \in [N]} \frac{1}{\cos^2\theta_\ell\left(\mathcal{E}_N^{\tilde{\mathcal{F}}}, \tilde{\mathcal{T}}(\boldsymbol{x})\right)} - 1 = \frac{\operatorname{Det}\tilde{\boldsymbol{K}}(\boldsymbol{x})}{\operatorname{Det}^2\boldsymbol{E}^{\tilde{\mathcal{F}}}(\boldsymbol{x})} - 1, \tag{158}$$

which has a tractable expectation under the projection DPP that we consider in this paper. As an illustration of (157), Figure 5 compares the quality of approximation of a mean element $\mu_g$ using kernel interpolation based on two configurations of nodes: the first configuration (top) is well spread and the second configuration (bottom) is not. Observe that the largest principal angle $\theta_N$ for the first configuration is around $\pi/4$, so that $\tan^2\theta_N \approx 1$; while it is around $\pi/2$ for the second configuration so that $\tan^2\theta_N \gg 1$. Now observe that the first design of nodes gives the best reconstruction. This observation is consistent with (157).

### E.2 The inclusion probability of DPPs and the Christoffel functions

The optimal distribution $q_\lambda$, see section 2.2, can be linked to the so-called *Christoffel functions* [19]. These functions are rooted in the literature on orthogonal polynomials [18]. To make it simpler, we introduce them in dimension $d = 1$. They are defined by

$$C_{\ell,\mathrm{d}\omega} : z \mapsto \min_{\substack{P \in \mathbb{R}_\ell[X] \\ P(z)=1}} \int_{\mathcal{X}} P(x)^2 \mathrm{d}\omega(x), \quad \ell \in \mathbb{N}. \tag{159}$$

Christoffel functions have a more explicit form [18] that can be used for pointwise evaluation

$$C_{\ell,\mathrm{d}\omega}(z) = \frac{1}{\sum_{m \leq \ell} P_m(z)^2}, \quad \ell \in \mathbb{N}, \tag{160}$$

where $(P_m)_{m \in \mathbb{N}}$ are the orthonormal polynomials with respect to $\mathrm{d}\omega$. To establish a connection with $q_\lambda$, the authors of [19] defined regularized Christoffel functions for some kernel $k$:

$$C_{\lambda,\mathrm{d}\omega,k} : z \mapsto \min_{\substack{f \in \mathcal{F} \\ f(z)=1}} \int_{\mathcal{X}} f(x)^2 \mathrm{d}\omega(x) + \lambda\|f\|_{\mathcal{F}}^2, \quad \lambda \in \mathbb{R}_+^*. \tag{161}$$

Figure 5: The dependency of the quality of reconstruction and the largest principal angle $\theta_N = \theta_N \left( \tilde{\mathcal{T}}(\boldsymbol{x}), \mathcal{E}_N^{\tilde{\mathcal{F}}} \right)$ for $N = 5$. A comparison of a design of nodes well-spread (above) and a design of nodes with clustering (below).

Figure 6: The inclusion probability of the projection DPP in the Gaussian case ($d = 1$): (a) the evaluations of the functions $x \mapsto \dfrac{\mathfrak{K}(x,x)}{N} \mathrm{d}\omega(x)$ for $N \in \{1, 2, 5, 10\}$ where $\mathrm{d}\omega$ is the measure of a normalized Gaussian variable, (b) the empirical inclusion probability based on 50000 realisations of the projection DPP compared to the evaluation of the Christoffel function ($N = 5$), the dots in red corresponds to the zeros of the scaled Hermite polynomial of order 5.

The authors derived an asymptotic equivalent of the function $C_{\lambda,w,k}$ in the regime $\lambda \to 0$ under some assumptions on the kernel. Furthermore, they proved that $C_{\lambda,w,k}$ is tied to $q_\lambda$ by the following relationship (Lemma 5, [19]):

$$q_\lambda(x) \propto \langle k(x, .), (\Sigma + \lambda \mathbb{I}_{\mathcal{H}})^{-1} k(x, .) \rangle_{\mathcal{F}} = \frac{1}{C_{\lambda,w,k}(x)}. \tag{162}$$

On the other hand, assume that the $(\psi_n)$ are the family of orthonormal polynomials with respect to $\mathrm{d}\omega$. Let $x \in \mathcal{X}$ and $\boldsymbol{x}$ a random subset of $\mathcal{X}^N$ drawn from the Projection DPP $(\mathfrak{K}, \mathrm{d}\omega)$, then

$$\mathbb{P}_{\mathrm{DPP}}(x \in \boldsymbol{x}) = \frac{1}{N} \mathfrak{K}(x, x) \mathrm{d}\omega(x) = \frac{1}{N} \sum_{n \in [N]} \psi_n(x) \psi_n(x) \mathrm{d}\omega(x) = \frac{1}{N C_{N,\mathrm{d}\omega}(x)} \mathrm{d}\omega(x). \tag{163}$$

In other words, the inclusion probability of the corresponding projection DPP is related to the inverse of the Christoffel function as defined in (160). Figure 6 illustrates the evaluations of the inclusion probability of the projection DPP in the case of RKHS defined by the Gaussian kernel along with the Gaussian measure in the real line. Recall that in this case the eigenfunctions are given by

$$\tilde{e}_m(.) = H_m(\sqrt{2c}.). \tag{164}$$

The theoretical analysis of the "bumps" of the functions $x \mapsto 1/N\, \mathfrak{K}(x,x) \mathrm{d}\omega(x)$ was carried out in [7]. More precisely, the authors studied the approximations of those bumps by Gaussians centred on the Hermite polynomials roots, see Figure 6 (b). We observe a similar behaviour for the multidimensional Gaussian case as illustrated in Figure 7: the inclusion probability of the projection DPP have has local maxima around the tensor products of the Hermite polynomials roots. In other words, the quadratures based on nodes sampled according to a projection DPP are probabilistic relaxations of classical quadratures based on roots of orthogonal polynomials that can be defined even if $N$ is not the square of an integer (the cases $N \in \{17, 21\}$ in Figure 7).

Figure 7: The inclusion probability of the projection DPP in the multidimensional Gaussian case $(d = 2)$: the evaluations of the functions $x \mapsto \dfrac{\mathfrak{K}(x, x)}{N} \otimes_{i=1}^{d} \mathrm{d}\omega(x_i)$ for $N \in \{16, 17, 21, 25\}$, the dots in black corresponds to the tensor product of the zeros of the scaled Hermite polynomials.

## Footnotes

[1]The matrix $\boldsymbol{A}_\ell$ depends on $\boldsymbol{x}$.