[Reviews · NeurIPS 2019]

Reviewer 1



*** UPDATE *** Thank you for your helpful response. It is clear that there are shortcomings in the presentation to be addressed, but on the basis that these will be addressed in a revision of the manuscript I am prepared to increase my score from 5 to 6. This paper considers kernel quadrature (KQ) methods for numerical integration. The novelty comes from taking a sample from a determinental point process (DPP) as the point set. The paper focuses on a theoretical analysis, and includes a cursory empirical check that the method performs as expected. An interesting finding is that DPP-KQ empirically out-performs Bayesian Monte Carlo (BMC). The theoretical contribution looks carefully written and technically accomplished. However, I am sorry to have to say that I do not think the main result is useful. I will try to explain why: Let's take the Sobolev space of order s on [0,1], as indeed the authors also consider in the manuscript. In that context, it has been shown that the minimal worst-case error (over the unit ball of the Sobolev space) for a quadrature method based on a deterministic point set is O(N^(-s)). For a randomly selected point set, the minimal mean square error (MSE) (over the unit ball) is O(N^(-2s-1)). I believe such results can be traced back to Bakvalov and Suld'in in the 1950s Soviet literature, but more modern accounts can be found in the information-based complexity literature of Novak and Wozniakowski in the 1980s. The authors do not appear to be aware of these results and, more importantly, they do not appear aware that the optimal O(N^(-s)) rate for a deterministic point set can easily be established for Bayesian Quadrature (BQ) by taking a uniform grid as the point set and applying the fill-distance-type bounds from e.g. the book "Scattered Data Approximation" of Holger Wendland. See e.g. [BOGOS2019,KOS2018] and the references therein for details. Such "uniform grids are optimal" results cover a range of settings and kernels, which rather raises the question of why one would want to use a random point set. Indeed, in the BQ work of O'Hagan the point set was selected to minimise the worst-case error (c.f. posterior standard deviation of the integral). Of course, BMC uses a random point set, and in some cases - such as integration on manifolds - a random set can be justified since it may not be possible to easily construct a grid. But in simple situations like integration on [0,1], I do not see why randomness would be helpful. The authors justify their analysis on several occasions by the claim that BMC does not have theoretical guarantees. In fact, a rate O(N^(-2s)) for the MSE was established for BMC in the Euclidean context in [BOGOS2019] and in the manifold context in [EGO2019]. This is actually a better rate than what the authors have demonstrated for DPP-KQ, and this rather undermines the extent of the contribution. All this being said, the authors demonstrate that DPP-KQ out-performs BMC. This is a nice finding, but it is not demonstrated extensively enough to justify the paper in its own terms. For example, the experiments were limited to dimension d = 1 and also g = 1. Minor: l23: Higher-order QMC methods also exist, which achieve optimal Sobolev rates - see the work of Josef Dick and colleagues. l24: Reference did not compile. l74. O'Hagan was not the inventor of BQ, that can be traced back at least to [L1972]. l76. The greedy selection of points in BQ can be theoretically analyses as a special case of the work of Ronald DeVore on greedy approximation in Hilbert spaces, so it is not true to say that there are no theoretical guarantees. l79. The authors state that "implementing it [BQ] usually requires some heuristics". This rather overlooks the substantial contribution of [JH2018], [KS2018], who reduced the computational cost of BQ to near linear in N. l103. \mathbb{N}^* undefined. l110. "speaking, seeing" -> "speaking, the probability of seeing" l144. It is not clear in the main text that k is being defined as thr limit of (10). l144. The property being assumed is sometimes called "unisolvency of the point set". l147. There is a Corollary 1 in the main text and the appendix, but they are different. l155. The use of "quadrature error" is not appropriate - this is not a quadrature error, since there is no integrand. More precisely, it is the worst case error over the unit ball of the RKHS. l178. The decreasing nature of the \sigma_n needs to be explicitly assumed. l195. The bound in (21) seems quite loose, especially if the eigenvalues are geometrically or exponentially decreasing? l224. The authors should take care that when they write "BQ" they really mean "BMC". Otherwise, they need to explicitly explain in the main text how the points for BQ are being selected. [BOGOS2019] Briol F-X, Oates, CJ, Girolami, M, Osborne, MA, Sejdinovic, D. Probabilistic Integration: A Role in Statistical Computation? (with discussion and rejoinder) Statistical Science, 34(1):1-22. (Rejoinder on p38-42.) [EGO2019] Ehler M, Gräf M, Oates CJ. Optimal Monte Carlo Integration on Closed Manifolds. Statistics and Computing, to appear, 2019. [JH2018] R. Jagadeeswaran and F. J. Hickernell. Fast automatic Bayesian cubature using lattice sampling, 2018. [KS2018] Karvonen, T. and Sarkka, S., 2018. Fully symmetric kernel quadrature. SIAM Journal on Scientific Computing, 40(2), pp.A697-A720. [KOS2018] Karvonen T, Oates CJ, Särkkä S. A Bayes-Sard Cubature Method. Advances in Neural Information Processing Systems (NeurIPS 2018). [L1972] Larkin, F. M. (1972). Gaussian measure in Hilbert space and applications in numerical analysis. Rocky Mountain J. Math., 2:(3) 379–421.

Reviewer 2



+ Detailed comments on the methodological and empirical contributions: The experiments lack a simple baseline of using uniform grids as design points of kernel quadrature. It is known that this method attains the optimal rate of convergence for deterministic quadrature in the Sobolev setting: see Corollary 1 of the following paper: Convergence Analysis of Deterministic Kernel-Based Quadrature Rules in Misspecified Settings https://link.springer.com/article/10.1007/s10208-018-09407-7 Thus, I'm wondering whether kernel quadrature with DPPs can outperform such a simple baseline. If not, what is the advantage of the proposed approach? An obvious drawback of the use of uniform grids is that it suffers from the curse of dimensionality. Thus, another question would be whether the use of DPPs works for for modestly large dimensional problems. This point might need a discussion. + References for Bayesian / kernel quadrature are not up-to-date. For instance, the following paper, which has been on arXiv for several years and gained a number of citations, is one of the key references for Bayesian quadrature. This paper provides convergence analysis of Bayesian quadrature methods (or equivalent kernel quadrature). The authors will also find other papers on Bayesian / kernel quadrature that appeared in machine learning conferences such as NeurIPS and ICML. Probabilistic Integration: A Role in Statistical Computation? Statist. Sci. Volume 34, Number 1 (2019), 1-22. https://projecteuclid.org/euclid.ss/1555056025 The following paper is strongly related to the topic of the current paper, and needs a discussion. On the Sampling Problem for Kernel Quadrature ICML 2017 http://proceedings.mlr.press/v70/briol17a.html + How the design points are generated for "BQ" in the experiments? Did the authors use the approach of Huszár and Duvenaud [17]? If so, this should be explicitly mentioned. The authors mention that BQ does not have theoretical guarantees, but this is a bit confusing. As shown in the above papers, there are several theoretical guarantees for BQ methods. Minor comments: - The authors mention that Proposition 1 is Proposition 2 in Bach [3], but it seems that this result Proposition 1 in [3]. Also, in Eq. (7), the supremum over the weights w seems to be infimum in the original result of [3]. - The notation should be defined. For instance, where is \mathbb{N}^* defined? It seems that this is the set of positive integers, but I don't think this notation is standard in the literature. - The Sobolev spaces discussed in this paper are periodic Sobolev spaces (also known as Korobov spaces in the QMC literature), so the authors should mention this.

Reviewer 3



Paper summary: motivated by the need to improve convergence rates for quadrature rule for functions living in an RKHS, the paper proposes to sample quadrature nodes from a determinantal point process (DPP), and the weights are found by solving a least-squares problem. The paper analyzes the expected squared error of the proposed quadrature rule, bounding the convergence rate in terms of the spectrum of the kernel. The paper then empirically validates the proposed method with numerical simulation for functions in RKHSs associated with the Sobolev and the Gaussian kernel, showing faster convergence than kernel herding and leverage score sampling. Strengths: 1. The idea of sampling from a DPPs, whose kernel is associated with the kernel of the RKHS, is interesting and novel to the best of my knowledge. 2. The proposed quadrature rule gets explicit convergence rate, for example in the case of finite N (number of nodes) when lambda = 0, unlike that of Bach [3]. 3. Numerical simulation shows that the proposed method performs wells, often on par with Bayesian quadrature (but with convergence rate) and better than Monte Carlo, kernel herding, and leverage score sampling [3]. 4. The bound in expectation using the DPP (Section 4.2.2) is elegant. Weaknesses: 1. The proposed method only gets convergence rate in expectation (i.e. only variance bound), not with high probability. Though Chebyshev's inequality gives bound in probability from the variance bound, this is still weaker than that of Bach [3]. 2. The method description lacks necessary details and intuition: - It's not clear how to get/estimate the mean element mu_g for different kernel spaces. - It's not clear how to sample from the DPP if the eigenfunctions e_n's are inaccessible (Eq (10) line 130). This seems to be the same problem with sampling from the leverage score in [3], so I'm not sure how sampling from the DPP is easier than sampling from the leverage score. - There is no intuition why DPP with that particular repulsion kernel is better than other sampling schemes. 3. The empirical results are not presented clearly: - In Figure 1: what is "quadrature error"? Is it the sup of error over all possible integrand f in the RKHS, or for a specific f? If it's the sup over all f, how does one get that quantity for other methods such as Bayesian quadrature (which doesn't have theoretical guarantee). If it's for a specific f, which function is it, and why is the error on that specific f representative of other functions? Other comment: - Eq (18), definition of principal angle: seems to be missing absolute value on the right hand side, as it could be negative. Minor: - Reference for Kernel herding is missing [?] - Line 205: Getting of the product -> Getting rid of the product - Please ensure correct capitalization in the references (e.g., [1] tsp -> TSP, [39] rkhss -> RKHSs) [3] F. Bach. On the equivalence between kernel quadrature rules and random feature expansions. The Journal of Machine Learning Research, 18(1):714–751, 2017. ===== Update after rebuttal: My questions have been adequately addressed. The main comparison in the paper seems to be the results of F. Bach [3]. Compared to [3], I do think the theoretical contribution (better convergence rate) is significant. However, as the other reviews pointed out, the theoretical comparison with Bayesian quadrature is lacking. The authors have agreed to address this. Therefore, I'm increasing my score.

[Author Response · NeurIPS 2019]

We thank the reviewers for their thorough and constructive reviews. As a general comment, we have now included the pointers given by R1 and R2 on general Bayesian quadrature (BQ), which we had indeed overlooked. In the manuscript, we meant « sequential BQ » as in (Huszar and Duvenaud, [20]). Using the suggested references, our approach is now introduced as randomized experimental designs for kernel quadrature, in the sense of Section 2.4.2 of (Briol et al., *Stat. Sci.* 2019); similarly, the DPPs used so far for numerical integration [5] are probabilistic relaxations of the classical Gaussian quadratures, themselves tightly connected to BQ (Karvonen and Särkkä, *MLSP* 2017). As requested by R2, we have also added deterministic grids and multivariate settings to our experiments, see Figure A and comments below.

R1: *optimal $\mathcal{O}(N^{-2s})$ rate for a deterministic point set can easily be established for BQ by taking a uniform grid [...] which rather raises the question of why one would want to use a random point set.*

Theorem 1 applies beyond the case of the uni-dimensional periodic Sobolev space, e.g., to Korobov spaces, to RKHSs on hyperspheres, to the space of band-limited functions restricted to an interval or to kernels defined over non-compact domains such as the Gaussian kernel on $\mathbb{R}^d$, etc. In particular, the last two examples correspond to exponentially decaying kernel eigenvalues. In these cases and unlike Sobolev, our bound is quite tight, as seen with the Gaussian kernel (see the notation paragraph for the general assumptions of Theorem 1). Beyond a new connection between DPPs and RHKSs, and as suggested by R1 and R2, we hope that the geometric arguments that we brought forward in our proofs can serve other approaches to BQ.

R2: *An obvious drawback of the use of uniform grids is that it suffers from the curse of dimensionality [...] whether the use of DPPs works for modestly large dimensional problems. This point might need a discussion*

We don't have conclusive theoretical arguments yet, but the way DPPs tie repulsive designs to the underlying RKHS may yield more meaningful bounds in $d > 1$ than fill-in distance arguments. In particular, we can expect explicit non-asymptotic bounds with smaller constants. Our manuscript shows that the key notion is the decay of the eigenvalues of the integration operator $\Sigma$. The dependence of that spectrum on the dimension can be explicitly worked out in the Sobolev and Korobov cases, and in general for tensor product of RKHSs; see Appendix A in [3]. However, what this says about DPP-KQ will have to wait for tighter bounds on the quadrature error, which may be tough nuts; see next bullet. Our manuscript is only a first brick in that wall.

R1: *A rate $\mathcal{O}(N^{-2s})$ for the MSE was established for BMC in [BOGOS2019]*

We now highlight this result in the manuscript. As commented above, our generic bound is indeed not as tight in the Sobolev case. Meanwhile, our experiments suggest that the rate $\mathcal{O}(N^{-2s})$ holds for DPP-KQ, and that the bound is representative of the behavior of the error even for small $N$, while Bach's LVSQ (with $\lambda = 0$) needs to wait for large values of $N$ for the error to actually fall down at that rate (see our Figure 1, and simulations in [3]). A potential way to tighten our bound when kernel eigenvalues decrease only polynomially, as in the Sobolev case, is discussed in Section 4.2. We are currently investigating this and trying to replace the term $N r_N$ by $r_N$ in Theorem 1. We even conjecture a bound that only involves eigenvalue $\sigma_{N+1}$. This is illustrated in the new experiment in Figure A. This figure also illustrates multivariate integration and a uniform grid as required by R2. The worst-case error of DPPKQ for $g \equiv 1$ (blue) scales as $\sigma_{N+1}$ (green), better than the sum $r_N$ of all eigenvalues above

Figure A: Example of an additional experiment for the case of a multivariate Korobov space, $d = 2$, $s = 1$ and $g \equiv 1$.

$\sigma_N$ (orange). We observed the same fast scaling for the Gaussian case when $d \geq 2$ (not shown). Such an improvement in our bound would propagate to all RKHSs; this generality is the strength of DPP-based experimental design.

R3: *not clear how to sample from the DPP if the eigenfunctions $e_n$'s are inaccessible (...) same problem as in [3]*

We stress that even when the eigenfunctions $e_n$ are accessible, it is not obvious how to sample from the regularized leverage-score distribution $q_\lambda^*$ of [3]; see our Eqn (6). Indeed, the RHS of (6) is an infinite sum. On the contrary, our projection DPP only involves eigenfunctions up to index $N$, and the conditionals in the chain rule can thus be computed. However, we agree that when the eigendecomposition of the kernel is not available, exact sampling from the DPP seems out of reach. We would then rely on MCMC like, e.g., (Chafai and Ferré, Arxiv:1806.05985).

R3: *There is no intuition why a DPP with that particular repulsion kernel is better than other sampling schemes.*

We have added both geometric and probabilistic intuition to the text. We sketch here the latter. First, it is natural to take a repulsion kernel $\mathfrak{K}$ that is tied to the RKHS kernel $k$: the smoother the integrand is in one area, the more repulsive the quadrature nodes can be in that area without contributing much quadrature error. Second, while it is theoretically possible to define a DPP with repulsion kernel proportional to $k$ [15], the resulting DPP would be a mixture of projection DPPs. The component with the highest weight in that mixture would be precisely the DPP we take in the paper. This intuitively suggests a variance reduction.

R3: *Explain the empirical results in Figure 1: what exactly is being plotted.*

For each number $N$ of design points and each method, we draw 50 independent designs $(x_i)$, compute the weights $(w_i)$, and we report the average of $\|\mu_g - \sum_1^N w_i k(x_i, \cdot)\|^2$. In both Sections 5.1 and 5.2, $\mu_g$ is available in closed form. We have now clarified this in the manuscript, and added implementation details about sampling.

[Meta-Review · NeurIPS 2019]

The paper proposes a kernel quadrature method with determinantal point processes, and provide some theoretical and experimental results. We think that the paper shows promising experimental results, but the obtained theoretical results may not be appropriately compared with the recent works on Bayesian quadrature. We recommend acceptance for this submission assuming that the final camera ready reflect the following points, which have been discussed in the Authors' Feedback. - Include references on Bayesian quadrature to reflect the recent advances, and discuss the relation with this work. - Discuss the known optimal rate for Bayesian quadrature including the one with uniform grids, and compare it with the current results. - Include the experimental results of Figure A and relevant discussions in the Authors' Feedback.